# A Modular Arbitrary-Order Ocean-Atmosphere Model: MAOOAM v1.0

Lesley De Cruz, Jonathan Demaeyer, and Stéphane Vannitsem

Royal Meteorological Institute of Belgium, Avenue Circulaire 3, 1180 Brussels, Belgium

*Correspondence to:* L. De Cruz (lesley.decruz@meteo.be)

**Abstract.**

This paper describes a reduced-order quasi-geostrophic coupled ocean-atmosphere model that allows for an arbitrary number of atmospheric and oceanic modes to be retained in the spectral decomposition. The modularity of this new model allows one to easily modify the model physics. Using this new model, coined "Modular Arbitrary-Order Ocean-Atmosphere Model" (MAOOAM), we analyse the dependence of the model dynamics on the truncation level of the spectral expansion, and unveil spurious behaviour that may exist at low resolution by a comparison with the higher resolution configurations. In particular, we assess the robustness of the coupled low-frequency variability when the number of modes is increased. An "optimal" configuration is proposed for which the ocean resolution is sufficiently high while the total number of modes is small enough to allow for a tractable and extensive analysis of the dynamics.

## 1 Introduction

The atmosphere at mid-latitudes displays a variability on a wide range of space and time scales, and in particular a low-frequency variability at interannual and decadal time scales as suggested by the analyses of different time series developed in past years (Trenberth, 1990; Trenberth and Hurrell, 1994; Hurrell, 1995; Mantua et al., 1997; Li and Wang, 2003; Lovejoy and Schertzer, 2013). In contrast to the phenomenon of El Niño – Southern Oscillation (ENSO), of which the driving mechanisms are intensively studied and quite well understood (e.g., Philander, 1990; Ghil and Zaliapin, 2013), the origin of mid-latitudes Low-Frequency Variability (LFV) remains highly debated, mainly due to the poor ability of state-of-the-art coupled ocean-atmosphere models to simulate it correctly (e.g., Nnamchi et al., 2011; Smith et al., 2014). The most plausible candidates of this LFV are either the coupling with the local ocean (Kravtsov et al., 2007) or teleconnections with the Tropical Pacific ocean-atmosphere variability (Müller et al., 2008), or both.

Recently the impact of the coupling between the ocean and the atmosphere at mid-latitudes on the atmospheric predictability (Nese and Dutton, 1993; Roebber, 1995; Peña and Kalnay, 2004) and the development of the LFV (van Veen, 2003) has been explored in a series of low-order coupled ocean-atmosphere systems. However, the limited flexibility of the possible geometries of these previous models led the present authors to develop a series of new model versions. The first of these, OA-QG-WS v1 (Vannitsem, 2014), for Ocean-Atmosphere - Quasi-Geostrophic - Wind Stress, features only mechanical coupling between the ocean and the atmosphere, and uses 12 atmospheric variables following Charney and Straus (1980) and four oceanic modes

following Pierini (2011). In a successor of this model, OA-QG-WS v2, the set of atmospheric variables is extended from 12 to 20 as in Reinhold and Pierrehumbert (1982). This increase in resolution in the atmosphere was shown to be key to the development of a realistic double gyre in the ocean (Vannitsem and De Cruz, 2014). A third version of this model, hereafter referred to as VDDG in reference to the authors of the model, includes passively advected temperature in the ocean and an energy balance scheme, combined with an extended set of modes for the ocean (Vannitsem et al., 2015).

In the VDDG model, an LFV associated with the coupling between the ocean and the atmosphere is successfully identified, allowing for extended-range coupled ocean-atmosphere predictions. Moreover the development of this coupled ocean-atmosphere mode is robust when stochastic forcings are added (Demaeyer and Vannitsem, 2016), or when a seasonal radiative forcing is incorporated in the low-order model (Vannitsem, 2015). Remarkably the presence of the seasonal radiative input favours the development of the coupled mode due to the amplification of the impact of the wind stress forcing in summer, associated with a drastic reduction of the mixed layer thickness at that period of the year. While these are encouraging results, which suggest the generic character of the coupled ocean-atmosphere mode, they need to be confirmed through the analysis of more sophisticated models, and in particular in higher-resolution coupled systems.

In this article, we present a model that generalizes the VDDG model by allowing for an arbitrary number of modes, or basis functions in which the dynamical fields are expanded. The modes can be selected independently for the ocean and the atmosphere, and for the zonal and meridional directions. The modular approach allows one to straightforwardly modify the model physics, such as changing the drag coefficient or adding a seasonal insolation. This model was coined MAOOAM: the Modular Arbitrary-Order Ocean-Atmosphere Model. The model equations and its technical implementation are detailed in Sect. 2. In Sect. 3, MAOOAM is used to investigate the dependence of the model dynamics, i.e. its climatology and the qualitative structure of its attractor, on the number of modes included. Furthermore, the development of the LFV as a function of the spectral truncation is discussed. Key results are summarized in Sect. 4.

## 2 Model formulation

The model is composed of a two-layer quasi-geostrophic (QG) atmosphere, coupled both thermally and mechanically to a QG shallow-water ocean layer, in the $\beta$-plane approximation. The atmospheric component is an extension of the QG model, first developed by Charney and Straus (1980) and further refined by Reinhold and Pierrehumbert (1982). The equations of motion for the atmospheric streamfunction fields $\psi_a^1$ at 250 hPa and $\psi_a^3$ at 750 hPa, and the vertical velocity $\omega = dp/dt$, read:

$$\frac{\partial}{\partial t}\left(\nabla^2 \psi_a^1\right) + J(\psi_a^1, \nabla^2 \psi_a^1) + \beta \frac{\partial \psi_a^1}{\partial x} = -k_d' \nabla^2(\psi_a^1 - \psi_a^3) + \frac{f_0}{\Delta p}\omega, \tag{1}$$

$$\frac{\partial}{\partial t}\left(\nabla^2 \psi_a^3\right) + J(\psi_a^3, \nabla^2 \psi_a^3) + \beta \frac{\partial \psi_a^3}{\partial x} = +k_d' \nabla^2(\psi_a^1 - \psi_a^3) - \frac{f_0}{\Delta p}\omega - k_d \nabla^2(\psi_a^3 - \psi_o). \tag{2}$$

The Coriolis parameter $f$ is linearized around a value $f_0$ estimated at latitude $\phi_0 = 45°$N, $f = f_0 + \beta y$, with $\beta = df/dy$. The parameters $k_d'$ and $k_d$ quantify the friction between the two atmospheric layers and between the ocean and the atmosphere, respectively, and $\Delta p = 500$ hPa is the pressure difference between the atmospheric layers.

The equation of motion for the streamfunction $\psi_o$ of the ocean layer reads (cfr. Pierini, 2011)):

$$\frac{\partial}{\partial t}\left(\nabla^2 \psi_o - \frac{\psi_o}{L_R^2}\right) + J(\psi_o, \nabla^2 \psi_o) + \beta \frac{\partial \psi_o}{\partial x} = -r\nabla^2 \psi_o + \frac{C}{\rho h}\nabla^2(\psi_a^3 - \psi_o). \tag{3}$$

$L_R$ is the reduced Rossby deformation radius, $\rho$ the density, $h$ the depth, and $r$ the friction at the bottom of the active ocean layer. The rightmost term represents the impact of the wind stress, and is modulated by the drag coefficient of the mechanical ocean-atmosphere coupling, $d = C/(\rho h)$.

The time evolution of the atmosphere and ocean temperatures $T_a$ and $T_o$ obeys the following equations:

$$\gamma_a \left(\frac{\partial T_a}{\partial t} + J(\psi_a, T_a) - \sigma \omega \frac{p}{R}\right) = -\lambda(T_a - T_o) + \epsilon_a \sigma_B T_o^4 - 2\epsilon_a \sigma_B T_a^4 + R_a \tag{4}$$

$$\gamma_o \left(\frac{\partial T_o}{\partial t} + J(\psi_o, T_o)\right) = -\lambda(T_o - T_a) - \sigma_B T_o^4 + \epsilon_a \sigma_B T_a^4 + R_o. \tag{5}$$

Here, $\gamma_a$ and $\gamma_o$ are the heat capacity of the atmosphere and the active ocean layer. $\psi_a = (\psi_a^1 + \psi_a^3)/2$ is the atmospheric barotropic streamfunction. $\lambda$ is the heat transfer coefficient at the ocean-atmosphere interface, and $\sigma$ is the static stability of the atmosphere, taken to be constant. The quartic terms represent the long-wave radiation fluxes between the ocean, the atmosphere and outer space, with $\epsilon_a$ the emissivity of the grey-body atmosphere and $\sigma_B$ the Stefan-Boltzmann constant. By decomposing the temperatures as $T_a = T_a^0 + \delta T_a$ and $T_o = T_o^0 + \delta T_o$, the quartic terms are linearized around equilibrium temperatures $T_a^0$ and $T_o^0$, as detailed in Appendix B of Vannitsem et al. (2015). $R_a$ and $R_o$ are the short-wave radiation fluxes entering the atmosphere and the ocean, that are also decomposed as $R_a = R_a^0 + \delta R_a$ and $R_o = R_o^0 + \delta R_o$.

The hydrostatic relation in pressure coordinates $(\partial \Phi/\partial p) = -1/\rho$ where the geopotential height $\Phi^i = f_0 \psi_a^i$ and the ideal gas relation $p = \rho R T$ allow one to write $\delta T_a = 2 f_0 \theta_a / R$, with $\theta_a \equiv (\psi_a^1 - \psi_a^3)/2$ often referred to as the baroclinic streamfunction. This can be used to eliminate the vertical velocity $\omega$ from Eqs. (1)–(2) and (4). This reduces the independent dynamical fields to the streamfunction fields $\psi_a$ and $\psi_o$, and the temperature anomalies $\delta T_a$ and $\delta T_o$.

The prognostic equations for these four fields are then non-dimensionalized by dividing time by $f_0^{-1}$, distance by a characteristic length scale $L$, pressure by the difference $\Delta p$, temperature by $f_0^2 L^2/R$ and the streamfunctions by $L^2 f_0$. A more detailed discussion of the model equations and their non-dimensionalization can be found in Vannitsem and De Cruz (2014); Vannitsem et al. (2015).

All the parameters of the model equations used in the present work are listed in Table 1.

## 2.1 Expansion of the dynamical fields

In non-dimensionalized coordinates $x' = x/L$ and $y' = y/L$, the domain is defined by $(0 \leq x' \leq \frac{2\pi}{n}, 0 \leq y' \leq \pi)$, with $n = 2L_y/L_x$ the aspect ratio between its meridional and zonal extent (see Table 1 for the value used here). The atmospheric flow is defined in a zonally periodic channel with no-flux boundary conditions in the meridional direction $(\partial \cdot_a /\partial x' \equiv 0$ at $y' = 0, \pi)$, whereas the oceanic flow is confined within an ocean basin by imposing no-flux boundaries in both meridional $(\partial \cdot_o /\partial x' \equiv 0$ at $y' = 0, \pi)$ and zonal $(\partial \cdot_o /\partial y' \equiv 0$ at $x' = 0, 2\pi/n)$ directions. These boundary conditions limit the functions used in the Fourier expansion of the dynamical fields. With the proper normalization, the basis functions for the atmosphere must be of

the following form, following the nomenclature of Cehelsky and Tung (1987):

$$F_P^A(x',y') = \sqrt{2}\cos(Py') \tag{6}$$

$$F_{M,P}^K(x',y') = 2\cos(Mnx')\sin(Py') \tag{7}$$

$$F_{H,P}^L(x',y') = 2\sin(Hnx')\sin(Py'). \tag{8}$$

Analogously, the oceanic basis functions must be of the form:

$$\phi_{H_o,P_o}(x',y') = 2\sin(\frac{H_o n}{2}x')\sin(P_o y'), \tag{9}$$

with integer values of $M, H, P, H_o, P_o$.

    For example, the spectral truncation used by Charney and Straus (1980) can be specified as Eqs. (6)–(8) with $M = H = 1$; $P \in \{1,2\}$. Reinhold and Pierrehumbert (1982) extend this set by two blocks of two functions each, and the resulting set can

be specified as $M, H \in \{1,2\}$; $P \in \{1,2\}$. The VDDG model has $M, H \in \{1,2\}$; $P \in \{1,2\}$ and $H_o \in \{1,2\}$; $P_o \in \{1,2,3,4\}$. Note that for consistency, the ranges for $M$ and $H$ should be the same. The distinction between $M$ and $H$ is, however, required to avoid ambiguities in the formulae of the inner products, as specified in Appendix A.

    For given ranges of $1 \le P \le P^{\max}$; $1 \le (M,H) \le H^{\max}$, and $1 \le P_o \le P_o^{\max}$; $1 \le H_o \le H_o^{\max}$, the number of basis functions can be calculated as:

$n_a = P^{\max}(2H^{\max} + 1);$                           $n_o = P_o^{\max} H_o^{\max}.$           (10)

    Ordering the basis functions as in Eqs (6)-(8), along increasing values of $M = H_{(o)}$ and then $P_{(o)}$, allows one to write the set as $\{F_i(x',y'), \phi_j(x',y')\}$ $(1 \le i \le n_a, 1 \le j \le n_o)$. The dynamical fields can then be written as the following truncated series expansions:

$$\psi_a(x',y',t) = \sum_{i=1}^{n_a} \psi_{a,i}(t)F_i(x',y') \tag{11}$$

$$\delta T_a(x',y',t) = \sum_{i=1}^{n_a} \delta T_{a,i}(t)F_i(x',y') = 2\frac{f_0}{R}\sum_{i=1}^{n_a}\theta_{a,i}(t)F_i(x',y') \tag{12}$$

$$\psi_o(x',y',t) = \sum_{j=1}^{n_o}\psi_{o,j}(t)(\phi_j(x',y') - \overline{\phi_j}) \tag{13}$$

$$\delta T_o(x',y',t) = \sum_{j=1}^{n_o}\delta T_{o,j}(t)\phi_j(x',y'). \tag{14}$$

    Furthermore, the short-wave radiation or insolation is determined by $\delta R_a = C_a F_1$; $\delta R_o = C_o F_1$. In Eq. (13), a term $\overline{\phi_j}$ is added to the oceanic basis function $\phi_j(x',y')$ in order to give it a vanishing spatial average. This is required to guarantee mass

conservation in the ocean (Cessi and Primeau, 2001; McWilliams, 1977), but otherwise does not affect the dynamics. Indeed,

it can be added *a posteriori* when plotting the field $\psi_o(x', y', t)$. This term is non-zero for odd $P_o$ and $H_o$,

$$\overline{\phi_j} = \frac{n}{2\pi^2} \int_0^\pi \int_0^{\frac{2\pi}{n}} \phi_j(x', y') dx' dy'$$

$$= 2\frac{((-1)^{H_o} - 1)((-1)^{P_o} - 1)}{H_o P_o \pi^2}. \tag{15}$$

The mass conservation is automatically satisfied for $\psi_a(x', y', t)$, as the spatial averages of the atmospheric basis functions $F_i(x', y')$ are zero.

Substituting the fields in Eqs. (1)–(5) and projecting on the different basis functions yields $2(n_a + n_o)$ ordinary differential equations (ODEs) for as many variables. Due to the linearization of the quartic temperature fields in Eqs. (4) and (5), these equations are at most bilinear (due to the advection term) in the variables $\psi_{a,i}, \theta_{a,i}, \psi_{o,j}$ and $\delta T_{o,j}$, which will henceforth jointly be referred to as $\eta_i$, the components of the state vector $\boldsymbol{\eta}$.

To construct the dynamical equations of these variables, one has to compute the various projections or inner products with the basis functions, for which the following shorthand notation will be used:

$$\langle S, G \rangle \equiv \frac{n}{2\pi^2} \int_0^\pi \int_0^{2\pi/n} dy' \, dx' \, S(x', y') \, G(x', y'). \tag{16}$$

As described by Cehelsky and Tung (1987), the inner products for the atmosphere can be computed as purely algebraic formulae of the wave numbers $P, M, H$. We reiterate these algebraic formulae in Sect. A1 of Appendix A, and extend them with the formulae for both the ocean-atmosphere coupling terms, and the ocean inner products in Sect. A2. The inner products can be represented as either 2D or 3D tensors, which are sparse, but generally not diagonal.

## 2.2 Technical implementation

Substituting the fields by Eqs. (11)-(14) and calculating the coefficients using the expressions for the inner products as in Appendix A yields a set of $N \equiv 2(n_a + n_o)$ prognostic ordinary differential equations. These equations are at most bilinear in the variables $\eta_i$ $(1 \leq i \leq N)$ due to the linearization of the radiative terms around a reference temperature present in Eqs. (4)-(5). This system of ODEs can therefore be most generically expressed as the sum of a constant, a matrix multiplication and a tensor contraction:

$$\frac{d\eta_i}{dt} = c_i + \sum_{j=1}^N m_{i,j} \, \eta_j + \sum_{j,k=1}^N t_{i,j,k} \, \eta_j \, \eta_k \qquad (1 \leq i \leq N). \tag{17}$$

This expression can be further simplified by adding a dummy variable that is identically equal to one: $\eta_0 \equiv 1$. This extra variable allows one to merge $c_i$, $m_{i,j}$ and $t_{i,j,k}$ into the tensor $\mathcal{T}_{i,j,k}$, in which the linear terms are represented by $\mathcal{T}_{i,j,0}$ and the constant term by $\mathcal{T}_{i,0,0}$:

$$\frac{d\eta_i}{dt} = \sum_{j=0}^N \sum_{k=0}^N \mathcal{T}_{i,j,k} \, \eta_j \, \eta_k \qquad (1 \leq i \leq N). \tag{18}$$

The elements of the tensor $\mathcal{T}_{i,j,k}$ are specified in Appendix B. Recasting the system of ordinary differential equations for $\eta_i$ in the form of a tensor contraction has certain advantages, as we will clarify below. The symmetry of Eq. (18) allows for a unique representation of $\mathcal{T}_{i,j,k}$, if it is taken to be upper triangular in the last two indices ($\mathcal{T}_{i,j,k} \equiv 0$ if $j > k$). Since $\mathcal{T}_{i,j,k}$ is known to be sparse, it is stored using the coordinate list representation, i.e. a list of tuples, $(i,j,k,\mathcal{T}_{i,j,k})$. This representation renders the computation of the tendencies $d\eta_i/dt$ computationally very efficient as well as conveniently parallelizable.

Two implementations of MAOOAM are provided as supplementary material: one in Lua and one in Fortran. The Lua code is optimized for LuaJIT, a just-in-time compiler for Lua (Pall, 2015), and runs about 20% slower than the Fortran version. By default, the model equations are numerically integrated using the Heun method. We have tested higher-accuracy methods, but these did not significantly change the results. The integration method can easily be changed; as an example, a fourth-order Runge-Kutta integrator is also included in the Lua implementation.

## 2.3 Derivation of Jacobian, tangent linear and adjoint models

The form of Eq. (18) allows one to easily compute the Jacobian matrix of this system of ODEs. Indeed, denoting the right hand side of Eq. (18) as $d\eta_i/dt = f_i$, the expression reduces to

$$J_{i,j} = \frac{df_i}{d\eta_j} = d\left(\sum_{k,l=0}^{N} \mathcal{T}_{i,k,l}\,\eta_k\,\eta_l\right)/d\eta_j \qquad\qquad (1 \le i,j \le N)$$

$$= \sum_{k=0}^{N} (\mathcal{T}_{i,k,j} + \mathcal{T}_{i,j,k})\,\eta_k. \qquad\qquad (19)$$

The differential form of the tangent linear (TL) model for a small perturbation $\boldsymbol{\delta\eta}^{TL}$ of a trajectory $\boldsymbol{\eta}^*$ is then simply (Kalnay, 2003)

$$\frac{d\delta\eta_i^{TL}}{dt} = \sum_{j=1}^{N} J_{i,j}^*\,\delta\eta_j^{TL} \qquad\qquad (1 \le i \le N)$$

$$= \sum_{j=1}^{N}\sum_{k=0}^{N} (\mathcal{T}_{i,k,j} + \mathcal{T}_{i,j,k})\,\eta_k^*\,\delta\eta_j^{TL}. \qquad\qquad (20)$$

To obtain the differential form of the adjoint model along the trajectory $\boldsymbol{\eta}^*$, the Jacobian is transposed to yield the following equations for the adjoint variable $\boldsymbol{\delta\eta}^{AD}$:

$$-\frac{d\delta\eta_i^{AD}}{dt} = \sum_{j=1}^{N} J_{j,i}^*\,\delta\eta_j^{AD} \qquad\qquad (1 \le i \le N)$$

$$= \sum_{j=1}^{N}\sum_{k=0}^{N} (\mathcal{T}_{j,k,i} + \mathcal{T}_{j,i,k})\,\eta_k^*\,\delta\eta_j^{AD}. \qquad\qquad (21)$$

## 3 Model dynamics

This section details some key results obtained with the model for various levels of spectral truncation, with the set of parameter values given in Table 1. The parameter values for $L$, $L_R$, $\lambda$, $r$, $d$, $C_o$, $C_a$, $k_d$ and $k'_d$ were selected as detailed in Vannitsem et al. (2015). The same value was chosen for $k_d$ and $k'_d$, as was done in Charney and Straus (1980), see also Vannitsem and De Cruz (2014). Unless otherwise stated, all the following results are obtained after first integrating the model for a transient period of 30726.5 years. The model is subsequently integrated for another 92179.6 years to obtain a sufficiently long trajectory from which good statistics can be extracted.

For the atmospheric part of the model, a previous study (Cehelsky and Tung, 1987), referred to as CT in the following, has shown that spurious chaos and a too large variability in the modes near the spectral cut-off could take place if the resolution is not high enough. These manifestations of spurious behaviour can lead to solutions that differ significantly from the solutions of the full partial differential equations (PDEs, here Eqs. (1)-(5)). These findings lead us to the important question of *convergence*: to what degree has the solution of the truncated equations converged towards the solution of the PDEs? Although we do not have access to the latter, one can infer how the solutions are altered when the resolution is increased. Therefore, it cannot be asserted that convergence has been reached, and this point was also clearly stated in CT. However, we can reasonably suppose that when the solutions stabilize, they give an insight into the full dynamics.

This question is now addressed for the coupled atmosphere-ocean model MAOOAM. Figures 1 and 2 display cross-sections of the attractors of the model for different resolutions. The three variables selected in this projection are $\psi_{a,1}$, $\psi_{o,2}$ and $\theta_{o,2}$, which have already been used to represent the large-scale variability of the model (Vannitsem et al., 2015). We use the same notation as in CT to specify the resolution of each component: $(H^{\max})x$-$(P^{\max})y$ for the atmosphere and $(H_o^{\max})x$-$(P_o^{\max})y$ for the ocean, with $M^{\max} = H^{\max}$. All the model configurations used are listed in Table 2. To alleviate the notation in the following, a model configuration denoted simply by $H^{\max}x$-$P^{\max}y$ indicates that the resolution is the same in both components: $H_o^{\max} = H^{\max}$ and $P_o^{\max} = P^{\max}$.

The first panel of Fig. 1, with the atm. $2x$-$2y$ oc. $2x$-$4y$ resolution, shows the typical attractor geometry found in Vannitsem et al. (2015) and Vannitsem (2015) with a noisy, seemingly periodic orbit associated with the development of a large low-frequency signal. However, as the resolution is increased in both the ocean and atmosphere component, this structure destabilizes and we obtain more compact, noisy attractors in Figs. 1 and 2. The cause of this structural change is an interesting question in itself, which is worth exploring further in the future as it is associated with the problem of structural stability of models, but is beyond the scope of the present work.

Regarding the question of convergence, the variability of the atmospheric variables becomes quite stable as the resolution increases beyond $6x$-$6y$. Indeed, the bounds of the attractors on the vertical axis ($\psi_{a,1}$) stabilize at this resolution. This result is in agreement with the findings of CT. On the other hand, the convergence is not yet reached for the oceanic variables whose variability is strongly affected by adding further modes as in the $7x$-$7y$ and the $8x$-$8y$ resolutions.

The impact of the resolution on the solutions can also be examined by computing the variance of each variable of the barotropic and baroclinic streamfunctions, since these are associated with the kinetic and potential energy of the system (Yao,

1980). The presence of spurious behaviour can then be detected through substantial changes in this variability. The distributions of the total variance of the variables $\psi_{a,i}$ and $\psi_{o,i}$ are depicted in Figs. 3–6. The results show that the variance distribution does not change much beyond the $4x$-$4y$ resolution for the atmospheric component. However, for the oceanic component, the variance distribution is strongly modified when the resolution increases and therefore one cannot conclude from Fig. 6 that

some sort of convergence is reached at the $8x$-$8y$ resolution. To interpret this specific property, one must recall an important feature of two-dimensional quasi-geostrophic turbulence, namely the presence of a specific space scale, the *Rhines scale*, which delimits the two regimes associated with a wave-dominated dynamics and a turbulent dynamics. This space scale is given by

$$L_{Rh} = \sqrt{\frac{U}{\beta}} \tag{22}$$

where $U$ represents the root-mean-square velocity of the energy containing scales (Rhines, 1975; Vallis and Maltrud, 1993;

Vallis, 2006) and $\beta = df/dy$ is the meridional derivative of the Coriolis parameter $f$. If one takes the typical velocity of the order of a few metres per second and a few centimetres per second within the atmosphere and the ocean at large scales, the typical length scales will be of the order of 1000 km and 100 km, respectively. Therefore the highest wave-numbers necessary to resolve the wave-dominated part within the atmosphere and the ocean differ by a factor of 10. Coming back to our analysis, if this limit is reached for the atmosphere in our model at $H, P = 4$–$5$, we should suspect that a value of $H_o/2 \approx P_o \approx 40$–$50$

should be used for the ocean. This of course imposes strong constraints on our reduced-order model and would considerably limit its flexibility.

Let us now focus on the development of the LFV in these different model configurations, and let us define the geopotential height difference $\delta z$ between the locations $(\pi/n, \pi/4)$ and $(\pi/n, 3\pi/4)$ of the model's non-dimensional domain:

$$\delta z(t) = z(\pi/n, \pi/4, t) - z(\pi/n, 3\pi/4, t) \qquad , \qquad z(x', y', t) = \frac{f_0}{g} \psi_a(x', y', t)$$

where $z$ is the geopotential height field, as in Vannitsem et al. (2015). The results shown in Figs. 7 and 8 indicate that the LFV, present for atm. $2x$-$2y$ oc. $2x$-$4y$ as in Vannitsem (2015), is a very weak signal at intermediate resolution, but develops again when the number of modes is increased, as shown by the 1-year and 5-years running means. It suggests that the LFV previously found in low-resolution versions (see Fig. 7, panel atm. $2x$-$2y$ oc. $2x$-$4y$) is a robust feature of the model. Moreover, at high resolution this LFV is weaker than for the VDDG model version, but it seems closer to the actual dynamics found for

the North-Atlantic Oscillation (NAO) as discussed in Li and Wang (2003) and Stephenson et al. (2000).

The climatologies of the atmospheric barotropic streamfunction expressed in geopotential height further highlight the changes in the statistical properties of the model as a function of resolution. As shown in Figs. 9 and 10, the convergence is pretty fast toward an averaged zonal atmospheric circulation as the model resolution is increased. By contrast, the convergence for the oceanic streamfunction $\psi_o$ is less clear (Figs. 11 and 12), although a recurrent "global" double gyre is present

for each resolution. As for the LFV, the topology of the gyres at high resolution and their small-scale structure also seem to depend on whether $H^{\max}$, $M^{\max}$, $H_o^{\max}$ and $P_o^{\max}$, $P^{\max}$ are even or odd numbers.

The previous results point toward the important question of the optimal resolution of the oceanic component needed to get a sufficiently low-resolution model, while keeping a dynamics with strong similarities with a very high-resolution model. To

answer this question, we have performed some higher resolution integrations, but on shorter time spans. The time span for each integration is given in Table 2.

The variance distributions of the oceanic streamfunction variables (see Fig. 13) have decreased at the spectral cut-off's edges compared to the distributions of the lower resolution model configurations shown on Fig. 5. However, this decrease is not sufficient and apparently spurious effects are still present. For instance, the decay is not identical in both directions, with a slower decay rate as the zonal wave-number $H_o$ increases. We can even notice a peak in the distribution around $H_0 = H_0^{\max}$ and $P_0 = 2$ for all these higher model resolutions. This indicates that in fact we are still far from a quantitatively representative solution in the ocean. It supports that, as stated previously, a resolution of the order of the Rhines scale is needed to achieve a good convergence. For the ocean, it corresponds to a 100 km resolution which would then require roughly 2000 modes. Such a model will of course be very computationally expensive and cannot be considered as a "reduced"-order model anymore.

However, the comparison between the atm. $5x$-$5y$ oc. $12x$-$12y$ model configuration and the $10x$-$10y$ or the $12x$-$12y$ model configuration shows that the former displays a large-scale behaviour close to the latter two, but with a reduced complexity and computational cost. This similarity can be assessed by considering the climatologies of these higher-resolution runs displayed in Fig. 14 and by watching the corresponding videos (see below). We therefore believe that the atm. $5x$-$5y$ oc. $12x$-$12y$ model configuration is a good candidate when investigating more realistic dynamics than the one presented in VDDG. It must however be stressed that the VDDG model is still an important tool in this hierarchy of models since it already contains the basic mechanisms leading to low-frequency variability. In addition, the climatologies shown in Fig. 14 confirm the dependence of the dynamics on whether $H^{\max}$, $M^{\max}$, $H_o^{\max}$ and $P_o^{\max}$, $P^{\max}$ are even or odd, and also the presence of a global double-gyre in the ocean.

Finally, the dynamics of the model for the various resolutions are also illustrated in the videos provided as supplementary material. These videos depict the time evolution of the streamfunction and temperature fields, as well as the geopotential height difference and the 3-dimensional phase-space projection shown in Figs. 1 and 2. They give an insight into the high-frequency atmospheric and low-frequency oceanic variability, and also show the interesting time evolution of the oceanic gyres. In these videos, a striking feature is the presence of a westward wave propagation within the ocean while the LFV is developing in the coupled system. This feature has been associated with the propagation of Rossby-like waves (Vannitsem, 2015).

## 4   Conclusions

A new reduced-order coupled ocean-atmosphere model is presented, extending the low-resolution versions previously published (Vannitsem and De Cruz, 2014; Vannitsem et al., 2015). It is referred to as MAOOAM for Modular Arbitrary-Order Ocean-Atmosphere Model. This new model retains the main features of the previous versions, but allows for the selection of an arbitrary resolution within the ocean and the atmosphere. Besides the potential utility of this new functionality for evaluating the impact of the number of modes on the dynamics (as it has been done here), it opens the possibility to address several new questions in a very flexible way, such as the development of a consistent stochastic parameterization scheme through

scales, the understanding of the predictability problem at multiple scales and the role of model error, or the implementation of a data-assimilation scheme for the coupled ocean-atmosphere system.

In the present work, we have studied the impact of the resolution on the model solution's dynamics, by investigating the properties of the attractors and the variance distributions in both the oceanic and atmospheric components. The conclusion that can be drawn is that the convergence of the atmospheric component of the system is quite fast (as noted in Cehelsky and Tung (1987)), with variance distributions decreasing rapidly as a function of scale. However, the convergence of the oceanic component is much slower. Consequently, none of the solutions presented so far have satisfactorily converged toward a dynamics that correctly reflects the wave-dominated regime of the coupled ocean-atmosphere system. This regime corresponds to a resolution associated with the Rhines scale (which for the ocean is equal to $100$ km, or equivalently, to wavenumbers of the order of $H_o^{\max}/2 \approx P_o^{\max} \approx 50$). This stresses the need of high-resolution oceanic models to correctly represent the full coupled dynamics. One coupled model configuration which could, however, be recommended so far is the atm. $5x$-$5y$ oc. $12x$-$12y$ configuration, which seems to display some robustness in the ocean climatology as compared to the full $10x$-$10y$ and $12x$-$12y$ configurations. This conclusion requires further investigation with even higher resolutions, together with the use of more advanced tools of analysis like the computation of the Lyapunov exponents as in Vannitsem and Lucarini (2016). These can be computed using the tangent linear model version for which an implementation is also provided. This will be the subject of a future investigation.

The robustness of the LFV pattern, one of the most interesting features of the model, has also been explored. As it turns out, a LFV is still present in a large portion of the model configurations explored (not in $2x$-$4y$, $3x$-$3y$ and $4x$-$4y$), but a weaker LFV signal is found when high-resolution configurations are used. A dominant signal is found with a wide variety of periods ranging from one to a hundred years, depending on the model configuration. A more detailed analysis of the underlying structure of the system's attractor is needed to clarify the origin of this diversity, for instance through a bifurcation analysis as in Vannitsem et al. (2015). Note that the VDDG model is still an important tool in this hierarchy of models, since it already contains the basic mechanisms leading to the LFV.

Another interesting finding is the change of structure of the climatologies of the ocean gyres when choosing even or odd wave numbers ($H^{\max}$, $M^{\max}$, $H_o^{\max}$ and $P_o^{\max}$, $P^{\max}$). Is this feature purely associated with the convergence toward a spatially continuous field, or does it reflect specific properties of the dynamical equations, such as symmetries or invariance? These questions are still open and will be the subject of a future investigation that should allow to clarify what is the best set of modes needed for the ocean description.

Finally, the aim of the model is to study the effects of specific physical interaction mechanisms between the ocean and the atmosphere on the mid-latitude climate, both at large and intermediate scales. The modular design of the code of the model is adapted to such purposes, with the possibility of implementing new components, such as oceanic active transport, time dependent forcings or salinity fields.

## 5   Code availability

MAOOAM v1.0 is freely available for research purposes in the supplementary material and is also available at http://github.com/Climdyn/MAOOAM. In addition, the code is archived at http://dx.doi.org/10.5281/zenodo.47507. A version of the Lua implementation which is parallelized using MPI is also available at http://github.com/Climdyn/MAOOAM/tree/mpi. The parallelized

5   version is archived at http://dx.doi.org/10.5281/zenodo.47510.

*Acknowledgements.*   This work is partly supported by the Belgian Federal Science Policy Office under contract BR/121/A2/STOCHCLIM. The figures and videos have been prepared with the Matplotlib software (Hunter, 2007).

## Appendix A: Formulae to compute the inner products

In the formulae of the inner products of the atmospheric modes, Cehelsky and Tung (1987) use the following helper functions:

$$B_1(u,v,w) = \frac{w+v}{u} \tag{A1}$$

$$B_2(u,v,w) = \frac{w-v}{u} \tag{A2}$$

$$\lambda(r) = 0 \ (r \ \text{even}) \ \text{or} \ 1 \ (r \ \text{odd}) \tag{A3}$$

$$S_1(u,v,w,z) = -\frac{1}{2}(zu + wv) \tag{A4}$$

$$S_2(u,v,w,z) = \frac{1}{2}(wv - zu) \tag{A5}$$

$$S_3(u,v,w,z) = -S_1(u,v,w,z) \tag{A6}$$

$$S_4(u,v,w,z) = S_2(u,v,w,z). \tag{A7}$$

The same notation will be used in this appendix. In what follows, $\delta_{ij}$ is the Kronecker delta, so that $\delta_{ij} = 1$ if $i = j$, and 0 otherwise. Likewise, the function $\delta(x)$ used in this appendix is defined as:

$$\delta(x) = \begin{cases} 1, & \text{if } x = 0 \\ 0, & \text{otherwise.} \end{cases} \tag{A8}$$

Using these functions, the various coefficients of the model are calculated, starting with the internal atmosphere coefficients.

### A1 Atmospheric coefficients

In the following, we consider the ordering of the basis function used in Eqs. (11)-(14). For the sake of clarity, we add an extra informative upper index specifying the type of the atmospheric function in the definitions below. However, the inner products are completely defined by the lower indices alone. The atmospheric functions are thus noted:

$$F_i^\alpha(x',y') = \begin{cases} \sqrt{2}\cos(P_i y') & \text{if} \quad \alpha = A \\ \cos(H_i n x')\sin(P_i y') & \text{if} \quad \alpha = K \\ \sin(M_i n x')\sin(P_i y') & \text{if} \quad \alpha = L \end{cases} \tag{A9}$$

and the oceanic functions:

$$\phi_i(x',y') = \sin(\frac{M_{o,i} n}{2} x')\sin(P_{o,i} y'). \tag{A10}$$

### A1.1 The $a_{i,j}$ coefficients

These coefficients correspond to the eigenvalues of the Laplacian operator acting on the spectral expansion basis functions:

$$a_{i,j}^{\alpha,\beta} = \left\langle F_i^\alpha, \nabla^2 F_j^\beta \right\rangle \quad , \quad \alpha,\beta \in \{A,K,L\} \tag{A11}$$

which are given for each case by:

$$a_{i,j}^{A,A} = -\delta_{ij}\, P_i^2 \tag{A12}$$

$$a_{i,j}^{K,K} = -\delta_{ij}\,(n^2 H_i^2 + P_i^2) \tag{A13}$$

$$a_{i,j}^{L,L} = -\delta_{ij}\,(n^2 M_i^2 + P_i^2). \tag{A14}$$

## 5    A1.2    The $c_{i,j}$ coefficients

These coefficients are needed to evaluate the contribution of the $\beta$-terms, and only involve the $K$- and $L$-type base functions.

$$c_{i,j}^{\alpha,\beta} = \left\langle F_i^\alpha, \partial_{x'} F_j^\beta \right\rangle \qquad , \quad \alpha,\beta \in \{K,L\}. \tag{A15}$$

We have that

$$c_{i,j}^{K,K} = c_{i,j}^{L,L} = 0 \tag{A16}$$

$$10 \quad c_{i,j}^{K,L} = M_i\,\delta(P_i - H_j)\,\delta(P_i - P_j) = -c_{j,i}^{L,K}. \tag{A17}$$

## A1.3    The $g_{i,j,k}$ coefficients

These coefficients are given by

$$g_{i,j,k}^{\alpha,\beta,\gamma} = \left\langle F_i^\alpha, J(F_j^\beta, F_k^\gamma) \right\rangle \qquad , \quad \alpha,\beta,\gamma \in \{A,K,L\} \tag{A18}$$

and the non-zero ones are given by:

$$g_{i,j,k}^{A,K,L} = -\frac{2\sqrt{2}}{\pi} M_j \left( \frac{B_1(P_i,P_j,P_k)^2}{B_1(P_i,P_j,P_k)^2 - 1} - \frac{B_2(P_i,P_j,P_k)^2}{B_2(P_i,P_j,P_k)^2 - 1} \right)$$

$$\times\, \delta(M_j - H_k)\, \lambda(P_i + P_j + P_k) \tag{A19}$$

$$g_{i,j,k}^{K,K,L} = S_1(P_j,P_k,M_j,H_k) \big\{ \delta(M_i - H_k - M_j)\,\delta(P_i - P_k + P_j)$$

$$- \delta(M_i - H_k - M_j)\,\delta(P_i + P_k - P_j)$$

$$+ \big[ \delta(H_k - M_j + M_i) + \delta(H_k - M_j - M_i) \big]\,\delta(P_k + P_j - P_i) \big\}$$

$$+ S_2(P_j,P_k,M_j,H_k) \big\{ \delta(M_i - H_k - M_j)\,\delta(P_i - P_k - P_j)$$

$$+ \big[ \delta(H_k - M_j - M_i) + \delta(M_i + H_k - M_j) \big]$$

$$\times \big[ \delta(P_i - P_k + P_j) - \delta(P_k - P_j + P_i) \big] \big\} \tag{A20}$$

$$g_{i,j,k}^{L,L,L} = S_3(P_j,P_k,H_j,H_k) \big\{ \delta(H_k + H_j - H_i)\,\delta(P_k - P_j + P_i)$$

$$+ \big[ \delta(H_k - H_j - H_i) - \delta(H_k - H_j + H_i) \big]\,\delta(P_k + P_j - P_i)$$

$$- \delta(H_k + H_j - H_i)\,\delta(P_k - P_j - P_i) \big\}$$

$$+ S_4(P_j,P_k,M_j,H_k) \big\{ \delta(H_k + H_j - H_i)\,\delta(P_k - P_j - P_i)$$

$$+ \big[ \delta(H_k - H_j + H_i) - \delta(H_k - H_j - H_i) \big]$$

$$\times \big[ \delta(P_k - P_j - P_i) - \delta(P_k - P_j + P_i) \big] \big\} \qquad \text{for} \quad k > j > i \tag{A21}$$

where we have used the functions defined at the beginning of this appendix. All the other permutations can be obtained thanks to

$$g_{i,j,k}^{\alpha,\beta,\gamma} = -g_{j,i,k}^{\beta,\alpha,\gamma} = g_{k,i,j}^{\gamma,\alpha,\beta} = g_{j,k,i}^{\beta,\gamma,\alpha}. \tag{A22}$$

### A1.4   The $b_{i,j,k}$ coefficients

These coefficients are given by

$$b_{i,j,k}^{\alpha,\beta,\gamma} = \langle F_i^\alpha, J(F_j^\beta, \nabla^2 F_k^\gamma) \rangle \qquad , \quad \alpha,\beta,\gamma \in \{K,L\}. \tag{A23}$$

Therefore we obtain :

$$b_{i,j,k}^{\alpha,\beta,\gamma} = a_{k,k}^{\gamma,\gamma} \langle F_i^\alpha, J(F_j^\beta, F_k^\gamma) \rangle = a_{k,k}^{\gamma,\gamma}\, g_{i,j,k}^{\alpha,\beta,\gamma}. \tag{A24}$$

### A1.5   The $s_{i,j}$ coefficients

These coefficients encode the inner products between the atmospheric and the oceanic basis functions:

$$s_{i,j}^\alpha = \langle F_i^\alpha, \phi_j \rangle \qquad , \quad \alpha \in \{A,K,L\} \tag{A25}$$

which gives:

$$s_{i,j}^A = 8\sqrt{2}\,P_{o,j}\,\frac{\lambda(H_{o,j})\,\lambda(P_{o,j}+P_i)}{\pi^2\,H_{o,j}\,(P_{o,j}^2-P_i^2)} \tag{A26}$$

$$s_{i,j}^K = 4\,H_{o,j}\,\frac{\lambda(2M_i+H_{o,j})\,\lambda(P_{o,j}-P_i)}{\pi\,H_{o,j}\,(H_{o,j}^2-4M_i^2)} \tag{A27}$$

$$s_{i,j}^L = \delta(P_{o,j}-P_i)\,\delta(2H_i-H_{o,j}). \tag{A28}$$

## 5  A1.6  The $d_{i,j}$ coefficients

These coefficients are related to the forcing of the ocean on the atmosphere. They are given by the formula

$$d_{i,j}^\alpha = \left\langle F_i^\alpha, \nabla^2\phi_j \right\rangle = M_{j,j}\,s_{i,j}^\alpha \qquad , \quad \alpha \in \{A,K,L\} \tag{A29}$$

where the $M_{j,j}$ are given by the eigenvalues of the Laplacian operator acting on the oceanic basis functions (see next section).

## A2  Oceanic coefficients

## 10  A2.1  The $K_{i,j}$ coefficients

These coefficients are related to the forcing of the atmosphere on the ocean. They are given by

$$K_{i,j}{}^\alpha = \left\langle \phi_i, \nabla^2 F_j^\alpha \right\rangle = a_{j,j}^{\alpha,\alpha}\,s_{j,i}^\alpha \qquad , \quad \alpha \in \{A,K,L\}. \tag{A30}$$

## A2.2  The $M_{i,j}$ coefficients

These coefficients identify with the eigenvalues of the Laplacian acting on the oceanic basis functions:

$$15 \quad M_{i,j} = \left\langle \phi_i, \nabla^2\phi_j \right\rangle = -\delta_{ij}(n^2 H_{o,i}^2/4 + P_{o,i}^2). \tag{A31}$$

## A2.3  The $N_{i,j}$ coefficients

These coefficients are needed to evaluate the contribution of the $\beta$-terms and are given by

$$N_{i,j} = \left\langle \phi_i, \partial_{x'}\phi_j \right\rangle = -2n\,H_{o,i}\,H_{o,j}\,\frac{\delta(P_{o,i}-P_{o,j})\,\lambda(H_{o,i}+H_{o,j})}{\pi\,(H_{o,j}^2-H_{o,j}^2)}. \tag{A32}$$

## A2.4  The $O_{i,j,k}$ coefficients

20  These coefficients are given by

$$O_{i,j,k} = \left\langle \phi_i, J(\phi_j,\phi_k) \right\rangle \tag{A33}$$

with

$$O_{i,j,k} = \frac{n}{2}\Big[ S_3(P_{o,j}, P_{o,k}, H_{o,j}, H_{o,k})$$
$$\times \big\{ [\delta(H_{o,k} - H_{o,j} - H_{o,i}) - \delta(H_{o,k} - H_{o,j} + H_{o,i}] \, \delta(P_{o,k} + P_{o,j} - P_{o,i})$$
$$+ \delta(H_{o,k} + H_{o,j} - H_{o,i}) \, [\delta(P_{o,k} - P_{o,j} + P_{o,i}) - \delta(P_{o,k} - P_{o,j} - P_{o,i})] \big\}$$
$$+ S_4(P_{o,j}, P_{o,k}, H_{o,j}, H_{o,k}) \big\{ [\delta(H_{o,k} + H_{o,j} - H_{o,i}) \, \delta(P_{o,k} - P_{o,j} - P_{o,i})]$$
$$+ [\delta(H_{o,k} - H_{o,j} + H_{o,i}) - \delta(H_{o,k} - H_{o,j} - H_{o,i})]$$
$$\times [\delta(P_{o,k} - P_{o,j} - P_{o,i}) - \delta(P_{o,k} - P_{o,j} + P_{o,i})] \big\} \Big]. \tag{A34}$$

## A2.5 The $C_{i,j,k}$ coefficients

These coefficients are given by

$$C_{i,j,k} = \big\langle \phi_i, J(\phi_j, \nabla^2 \phi_k) \big\rangle = M_{k,k} \, O_{i,j,k}. \tag{A35}$$

## A2.6 The $W_{i,j}$ coefficients

These coefficients are related to the short-wave radiative forcing of the ocean and are given by

$$W_{i,j}{}^\alpha = \big\langle \phi_i, F_j^\alpha \big\rangle = s_{j,i}^\alpha \qquad , \quad \alpha \in \{A, K, L\}. \tag{A36}$$

## Appendix B: Definition of the tensor $\mathcal{T}_{i,j,k}$

The system of non-dimensionalized ODEs for the model variables is encoded in the model tensor $\mathcal{T}_{i,j,k}$, of which the complete definition is given in this appendix. To alleviate the notations, we use a shorthand notation for the indices of the different variables,

$$\psi_i = i \qquad\qquad\qquad\qquad (1 \le i \le n_a)$$
$$\theta_i = i + n_a \qquad\qquad\qquad\quad (1 \le i \le n_a)$$
$$\Psi_i = i + 2n_a \qquad\qquad\qquad\quad (1 \le i \le n_o)$$
$$\Theta_i = i + 2n_a + n_o \qquad\qquad\quad (1 \le i \le n_o). \tag{B1}$$

Furthermore, we suppress the upper indices which indicate the atmospheric function types, but are otherwise not needed to unambiguously specify the inner products.

## B1 Atmosphere equations

The components of the tensor for the atmosphere streamfunction are given by:

$$\mathcal{T}_{\psi_i,\psi_j,0} = -\frac{c_{i,j}\,\beta'}{a_{i,i}} - \frac{k_d}{2}\delta_{i,j}$$

$$\mathcal{T}_{\psi_i,\theta_j,0} = \frac{k_d}{2}\delta_{i,j}$$

$$\mathcal{T}_{\psi_i,\psi_j,\psi_k} = \mathcal{T}_{\psi_i,\theta_j,\theta_k} = -\frac{b_{i,j,k}}{a_{i,i}}$$

$$\mathcal{T}_{\psi_i,\Psi_j,0} = \frac{k_d\,d_{i,j}}{2a_{i,i}}. \tag{B2}$$

The atmospheric temperature equations are determined by the tensor elements:

$$\mathcal{T}_{\theta_1,0,0} = \frac{C_a'}{1 - a_{1,1}\,\sigma_0}$$

$$\mathcal{T}_{\theta_i,\psi_j,0} = \frac{a_{i,j}\,k_d\,\sigma_0}{2a_{i,i}\,\sigma_0 - 2}$$

$$\mathcal{T}_{\theta_i,\theta_j,0} = -\frac{\sigma_0\,(2\,c_{i,j}\,\beta' + a_{i,j}(k_d + 4\,k_d')) + 2\,(S_{B,a}' + \lambda_a')\delta_{i,j}}{2\,a_{i,i}\,\sigma_0 - 2}$$

$$\mathcal{T}_{\theta_i,\psi_j,\theta_k} = \frac{g_{i,j,k} - b_{i,j,k}\,\sigma_0}{a_{i,i}\,\sigma_0 - 1}$$

$$\mathcal{T}_{\theta_i,\theta_j,\psi_k} = \frac{b_{i,j,k}\,\sigma_0}{a_{i,i}\,\sigma_0 - 1}$$

$$\mathcal{T}_{\theta_i,\Psi_j,0} = \frac{k_d\,d_{i,j}\,\sigma_0}{2 - 2a_{i,i}\,\sigma_0}$$

$$\mathcal{T}_{\theta_i,\Theta_j,0} = s_{i,j}\frac{2\,S_{B,o}' + \lambda_a'}{2 - 2\,a_{i,j}\,\sigma_0} \tag{B3}$$

where we used the non-dimenionalized quantities

$$\beta' = \beta L/f_0$$

$$C_a' = C_a R/(2\gamma_a f_0^3 L^2)$$

$$\sigma_0 = \sigma \Delta p^2/(2L^2 f_0^2)$$

$$\lambda_a' = \lambda/(\gamma_a f_0)$$

$$S_{B,o}' = 2\epsilon_a \sigma_B \left(T_o^0\right)^3/(\gamma_a f_0)$$

$$S_{B,a}' = 8\epsilon_a \sigma_B \left(T_a^0\right)^3/(\gamma_a f_0).$$

## B2 Ocean equations

The components of the tensor for the ocean streamfunction are:

$$\mathcal{T}_{\Psi_i,\psi_j,0} = -\mathcal{T}_{\Psi_i,\theta_j,0} = \frac{K_{i,j}\, d'}{M_{i,i} + \gamma}$$

$$\mathcal{T}_{\Psi_i,\Psi_j,0} = -\frac{N_{i,j}\, \beta' + \delta_{i,j}\, M_{i,i}\, (r' + d')}{M_{i,i} + \gamma}$$

$$\mathcal{T}_{\Psi_i,\Psi_j,\Psi_k} = -\frac{C_{i,j,k}}{M_{i,i} + \gamma},\tag{B4}$$

with $\gamma = -L/L_R$, $d' = d/f_0$ and $r' = r/f_0$.

Finally, the equations for the ocean temperature are determined by:

$$\mathcal{T}_{\Theta_i,0,0} = C'_o\, W_{i,1}$$

$$\mathcal{T}_{\Theta_i,\theta_j,0} = W_{i,j}(2\,\lambda'_o + \sigma'_{B,a})$$

$$\mathcal{T}_{\Theta_i,\Theta_j,0} = -\delta_{i,j}(\lambda'_o + \sigma'_{B,o})$$

$$\mathcal{T}_{\Theta_i,\Psi_j,\Theta_k} = -O_{i,j,k}\tag{B5}$$

where the following non-dimensionalized quantities are used:

$$C'_o = C_o R/(\gamma_a f_0^3 L^2)$$

$$\lambda'_o = \lambda/(\gamma_o f_0)$$

$$\sigma'_{B,o} = 4\sigma_B \left(T_o^0\right)^3/(\gamma_o f_0)$$

$$\sigma'_{B,a} = 8\epsilon_a \sigma_B \left(T_a^0\right)^3/(\gamma_o f_0).$$

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

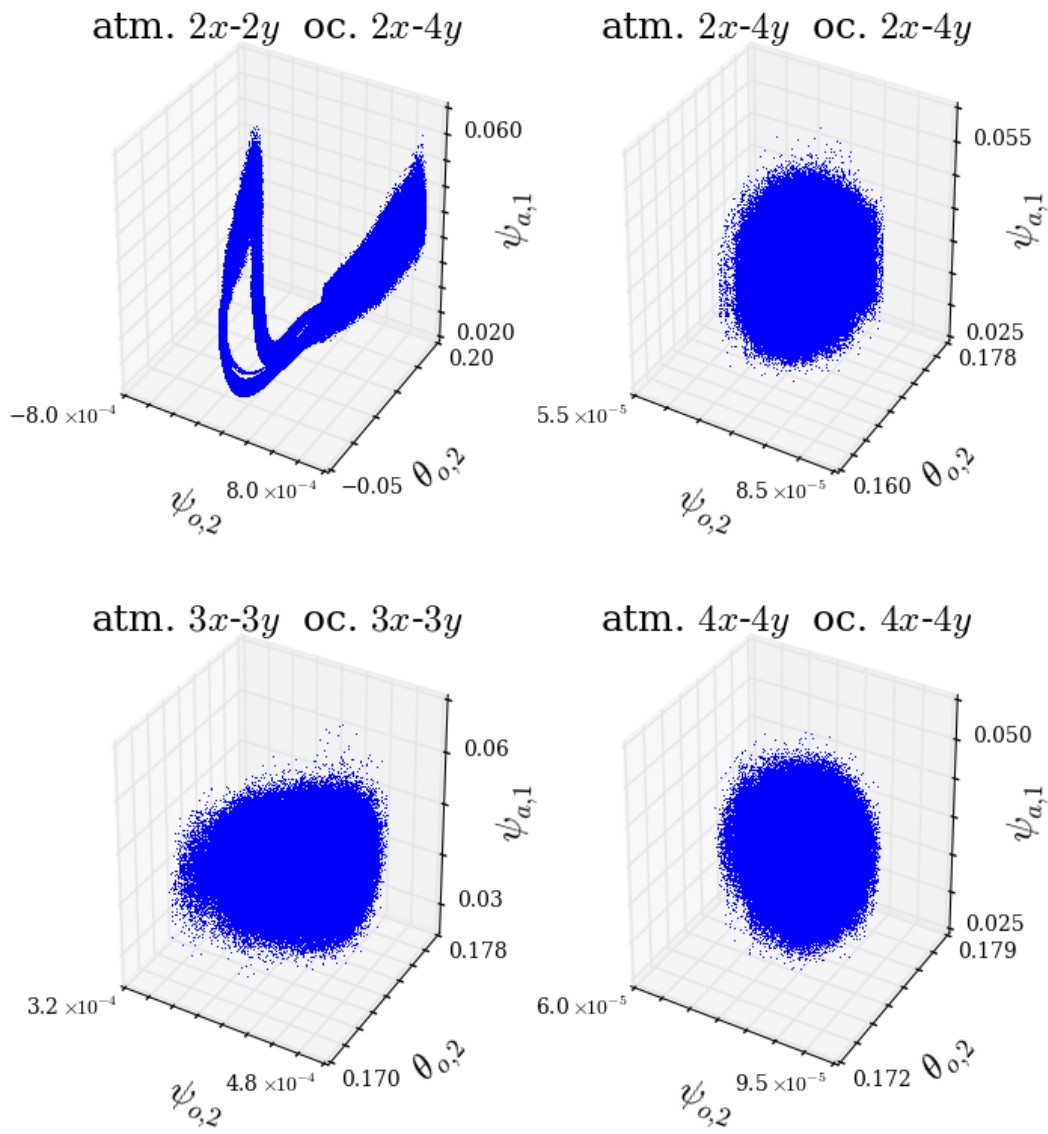

**Figure 1.** Cross-section of the attractors for various model resolutions. The atmospheric and oceanic resolution are both indicated above each panel. The parameters are given in Table 1.

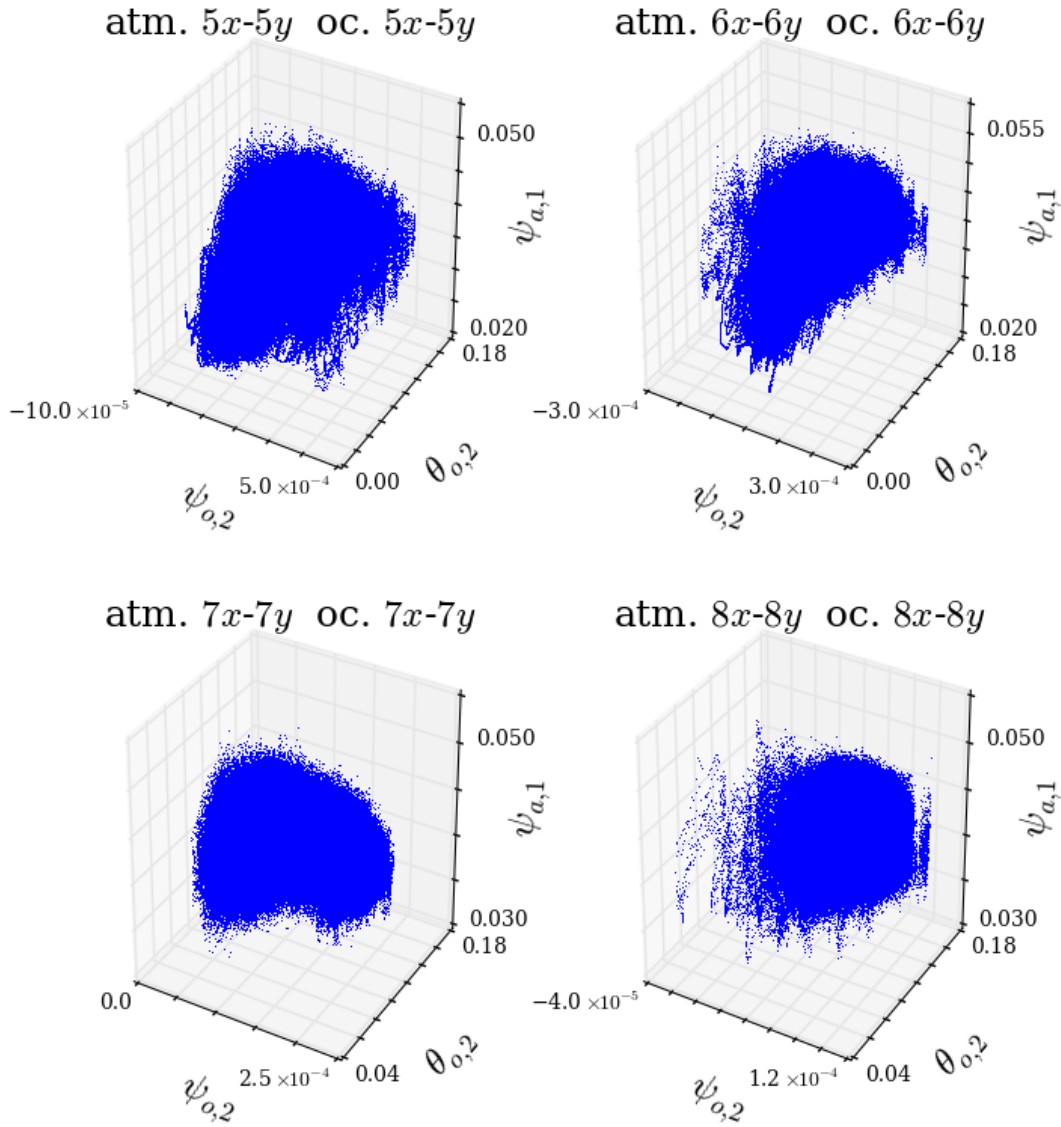

**Figure 2.** Cross-section of the attractors for various model resolutions (continued from Fig. 1).

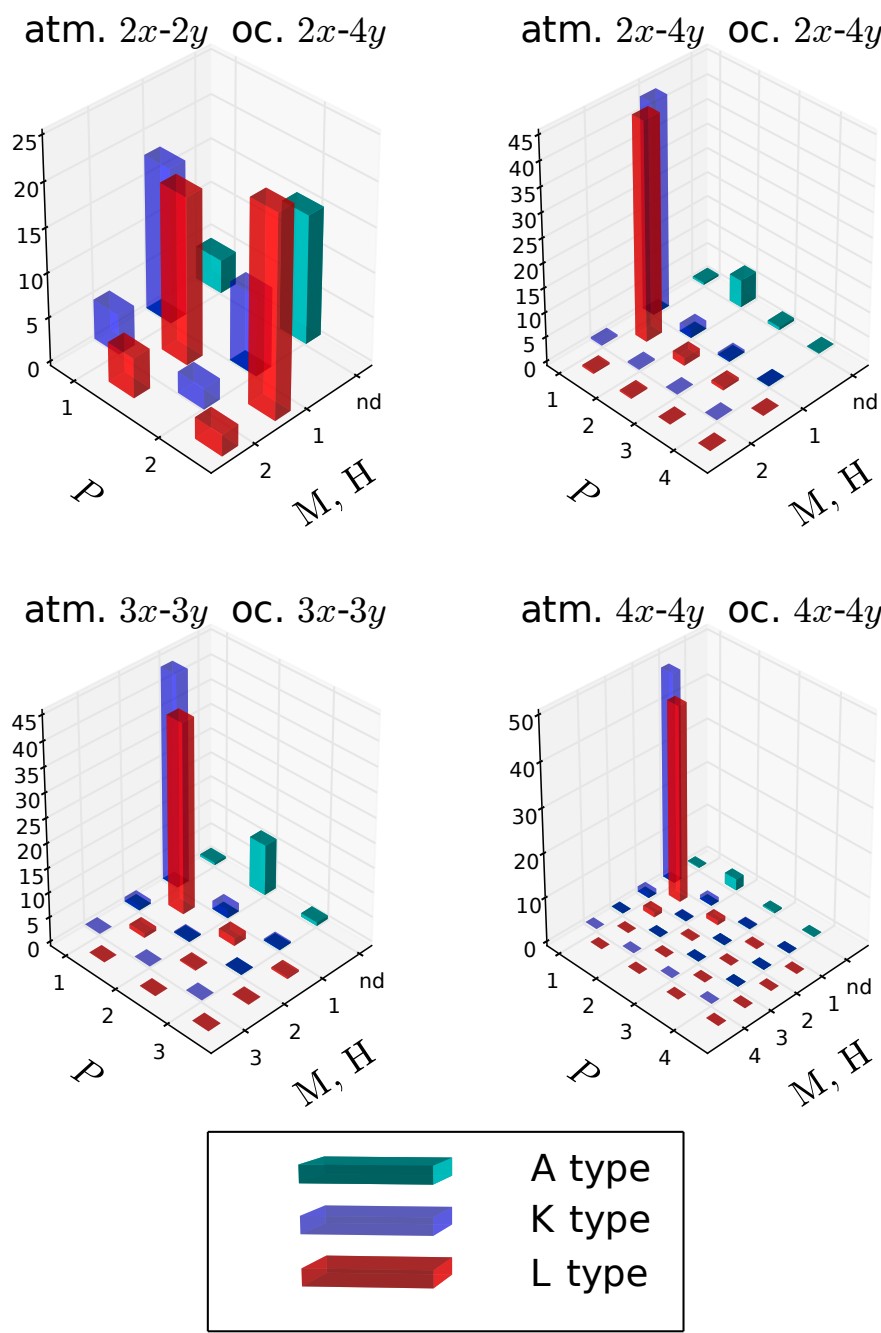

**Figure 3.** Variance distributions of the $\psi_{a,i}$ variables in percents for various model resolutions. For the variables associated with the $A$-type basis functions, the wavenumbers $M$ and $H$ are not defined (nd).

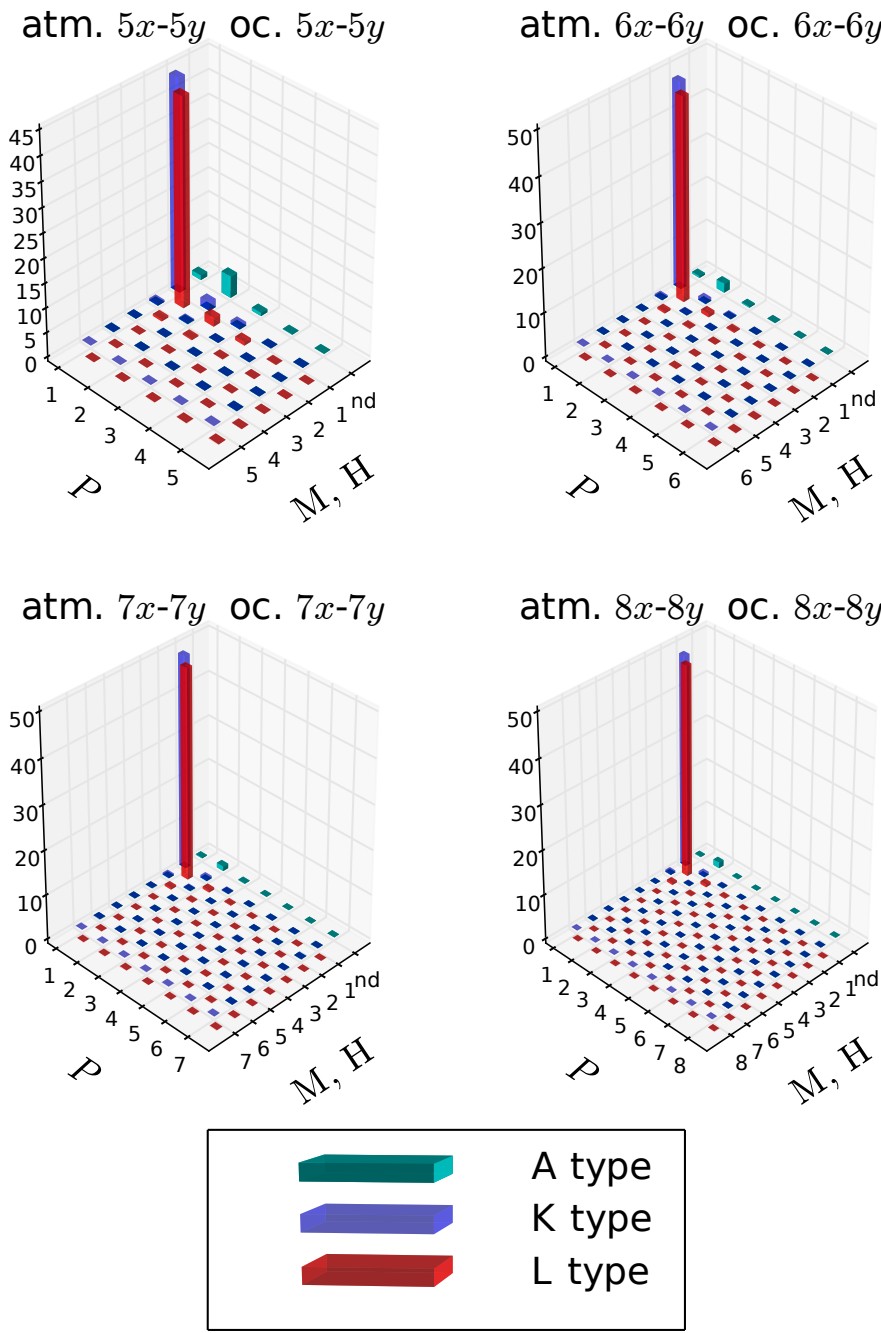

**Figure 4.** Variance distributions of the $\psi_{a,i}$ variables in percents for various model resolutions (continued from Fig. 3). For the variables associated with the $A$-type basis functions, the wavenumbers $M$ and $H$ are not defined (nd).

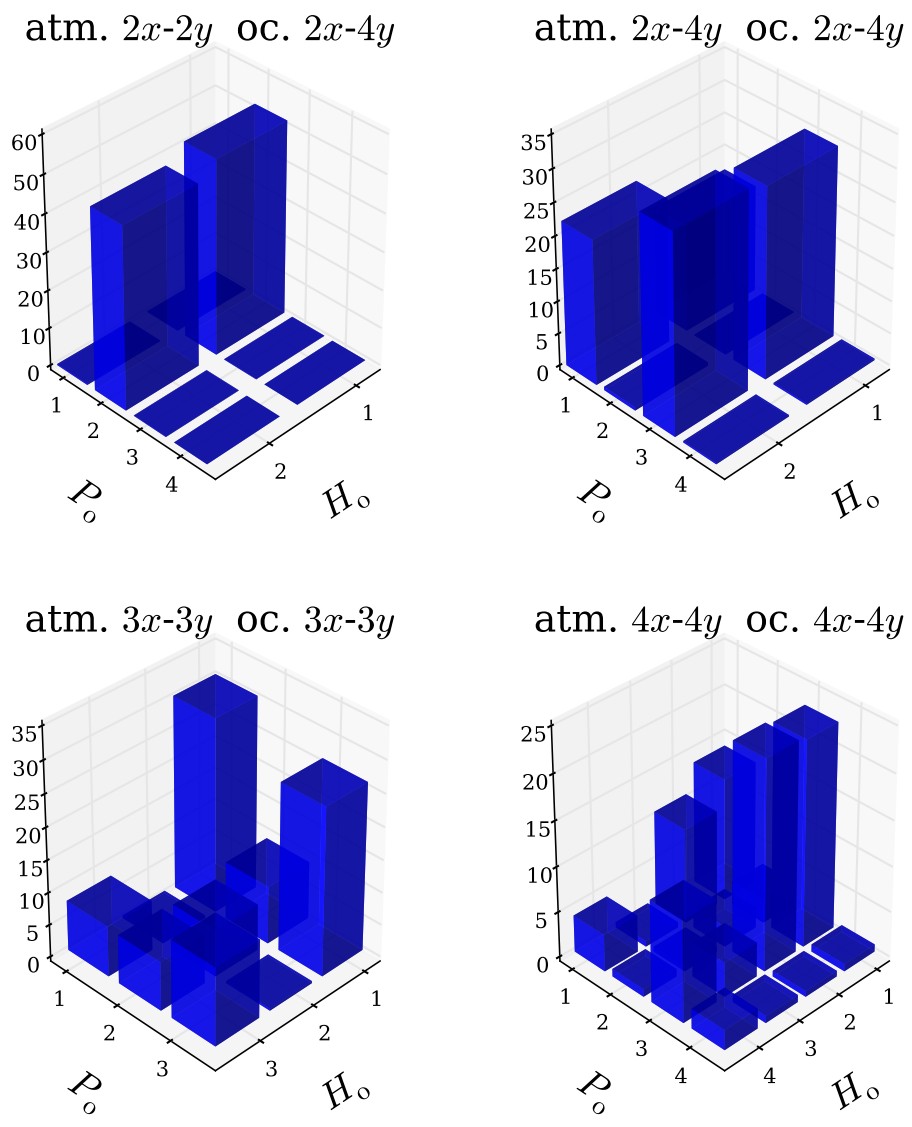

**Figure 5.** Variance distributions of the $\psi_{o,i}$ variables in percents for various model resolutions.

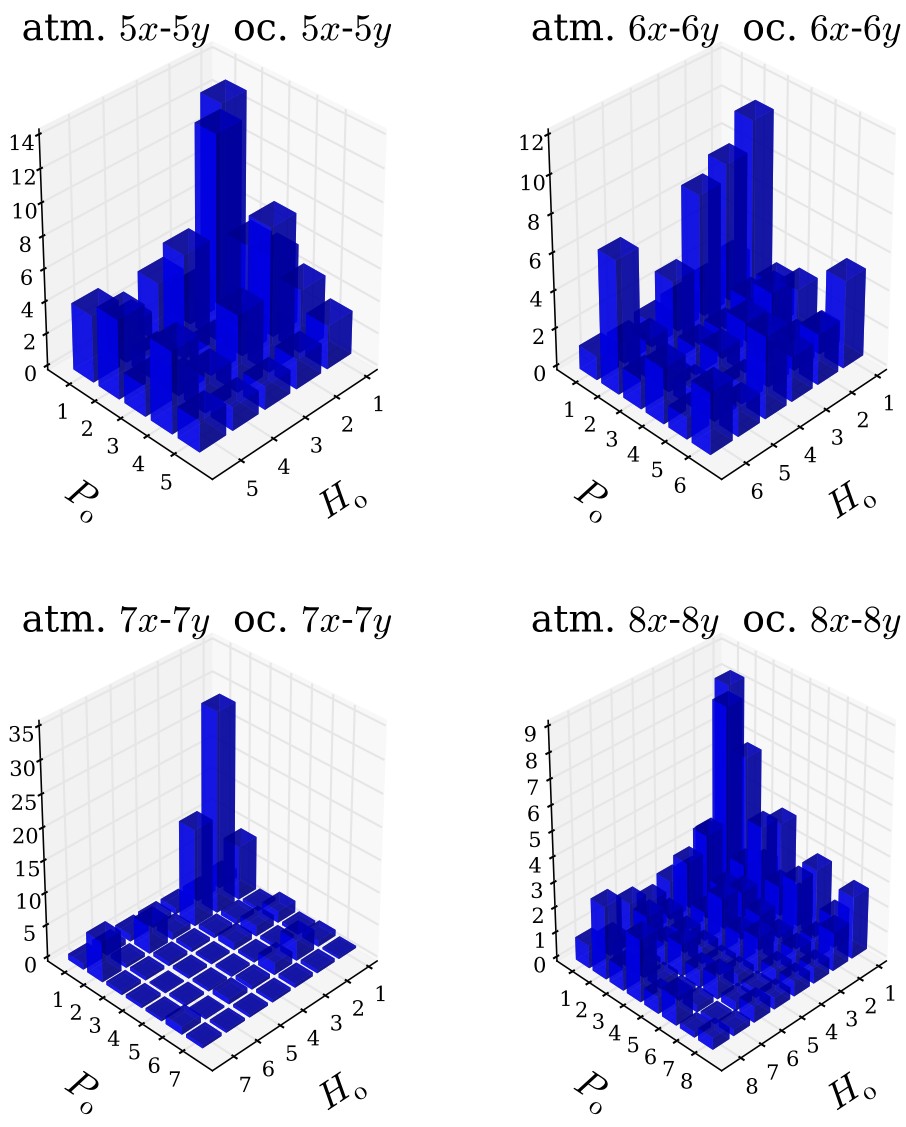

**Figure 6.** Variance distributions of the $\psi_{o,i}$ variables in percents for various model resolutions (continued from Fig. 5).

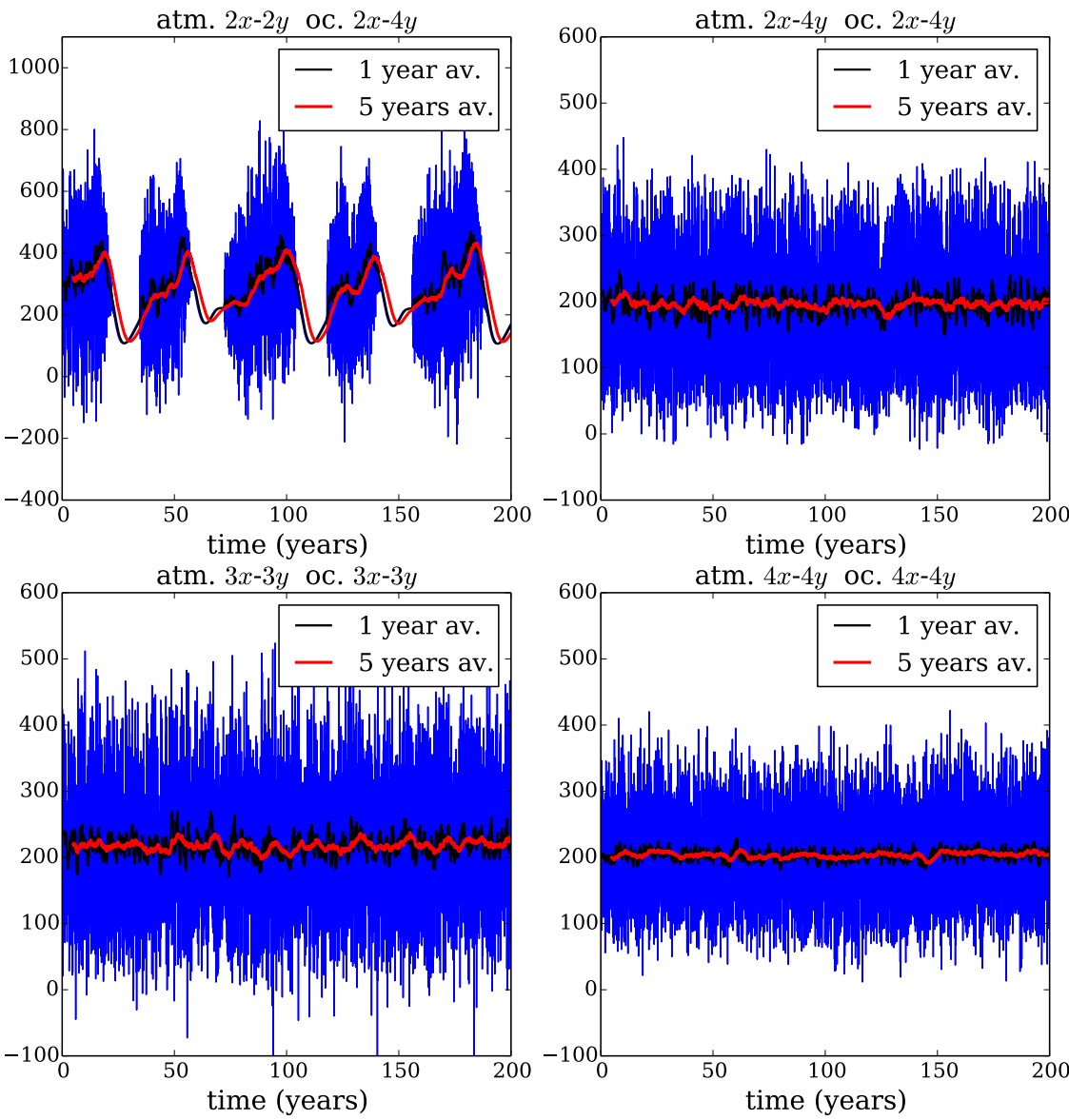

**Figure 7.** Time series of the geopotential height difference (m) between locations $(\pi/n, \pi/4))$ and $(\pi/n, 3\pi/4))$ of the model's non-dimensional domain for different resolution. Running averages for $\tau = 1$ y (black) and $\tau = 5$ y (red) are also provided, highlighting the LFV signal present in the series.

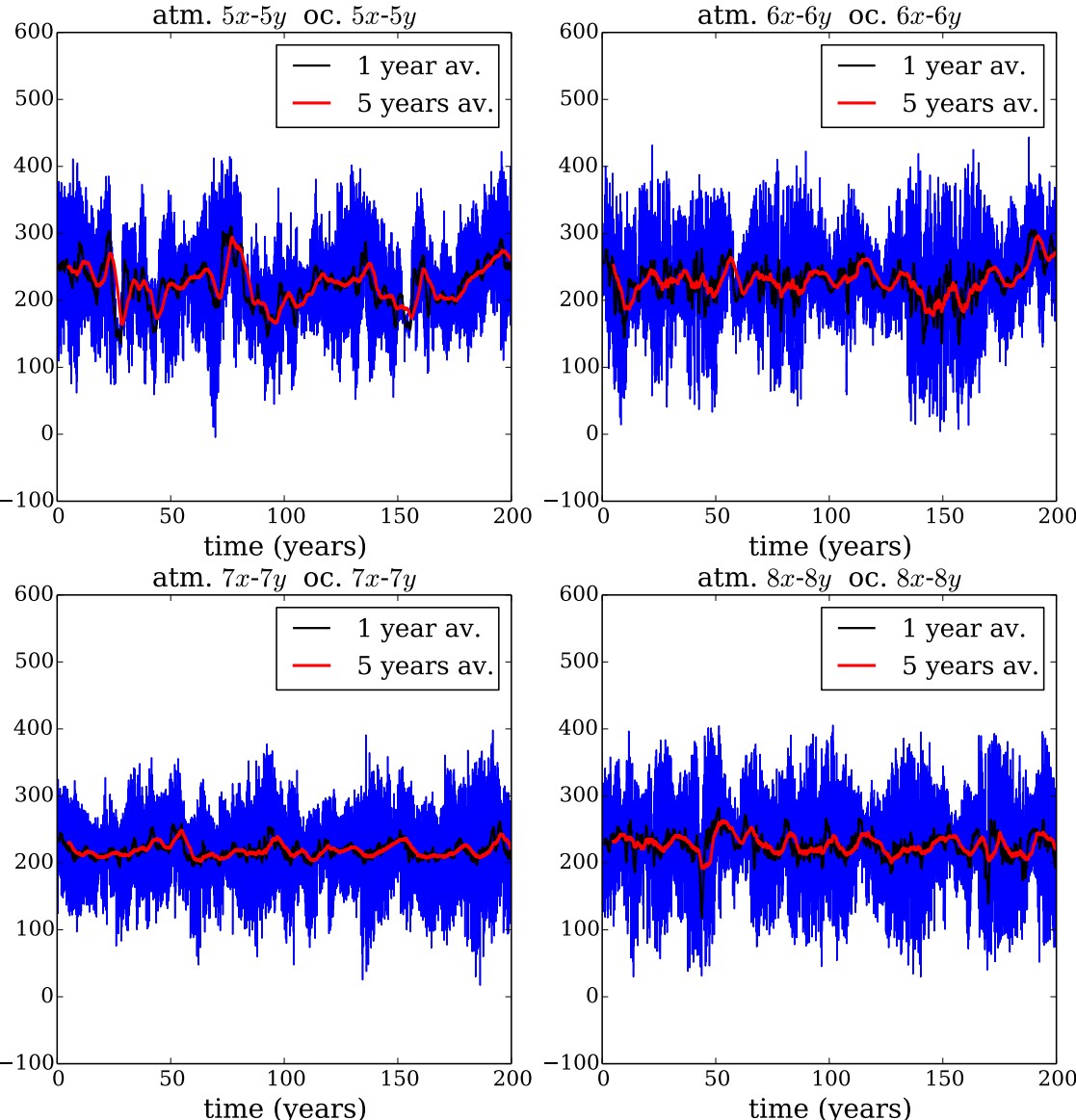

**Figure 8.** Time series of the geopotential height difference (continued from Fig. 7).

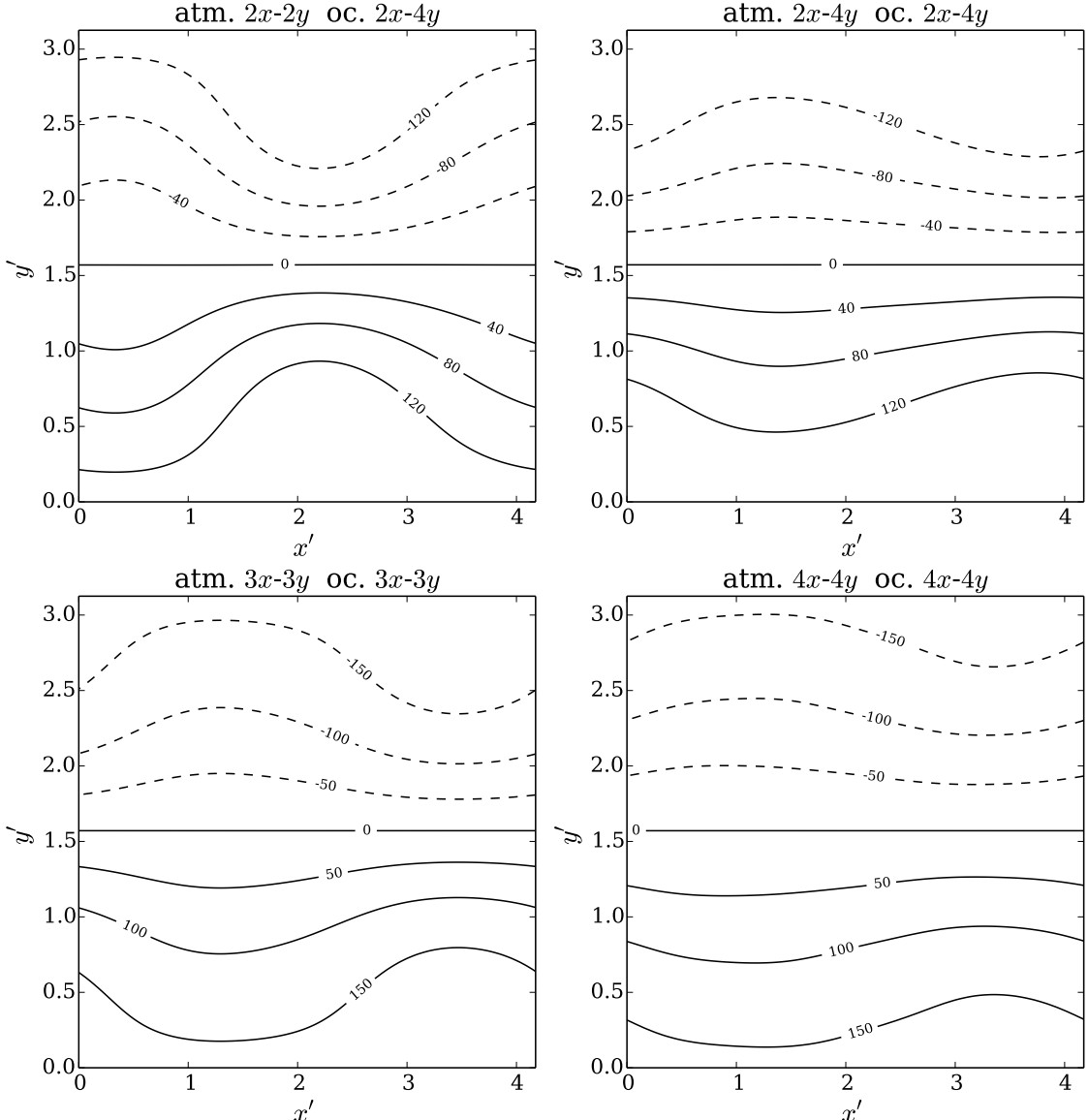

**Figure 9.** Climatologies for the geopotential height field $z = \frac{f_0}{g}\, \psi_a$ (m) presented on the non-dimensional model domain, as obtained using 92179.6 years of model integration.

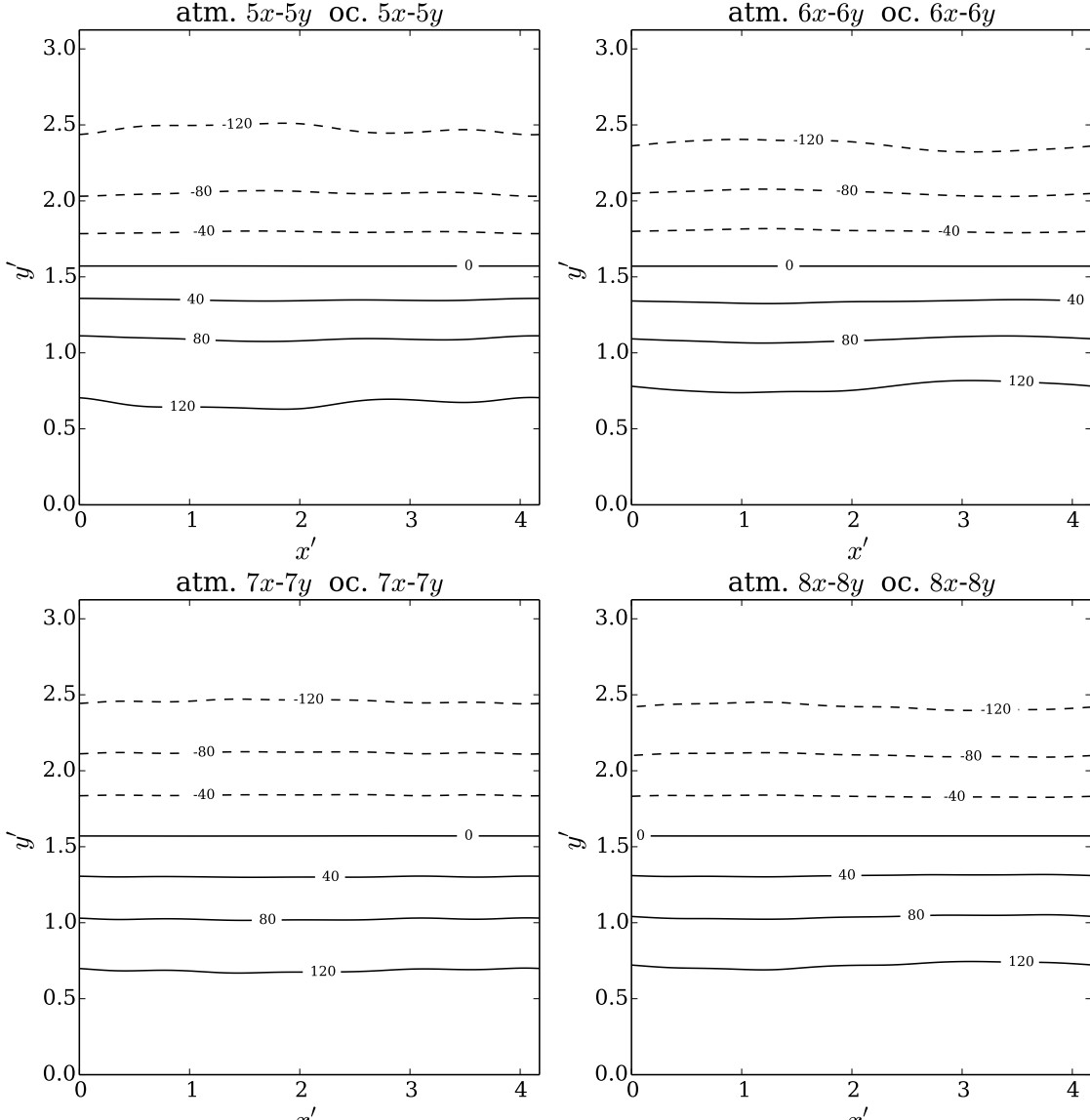

**Figure 10.** Climatologies for the geopotential height field $z = \frac{f_0}{g}\,\psi_a$ (m) (continued from Fig. 9).

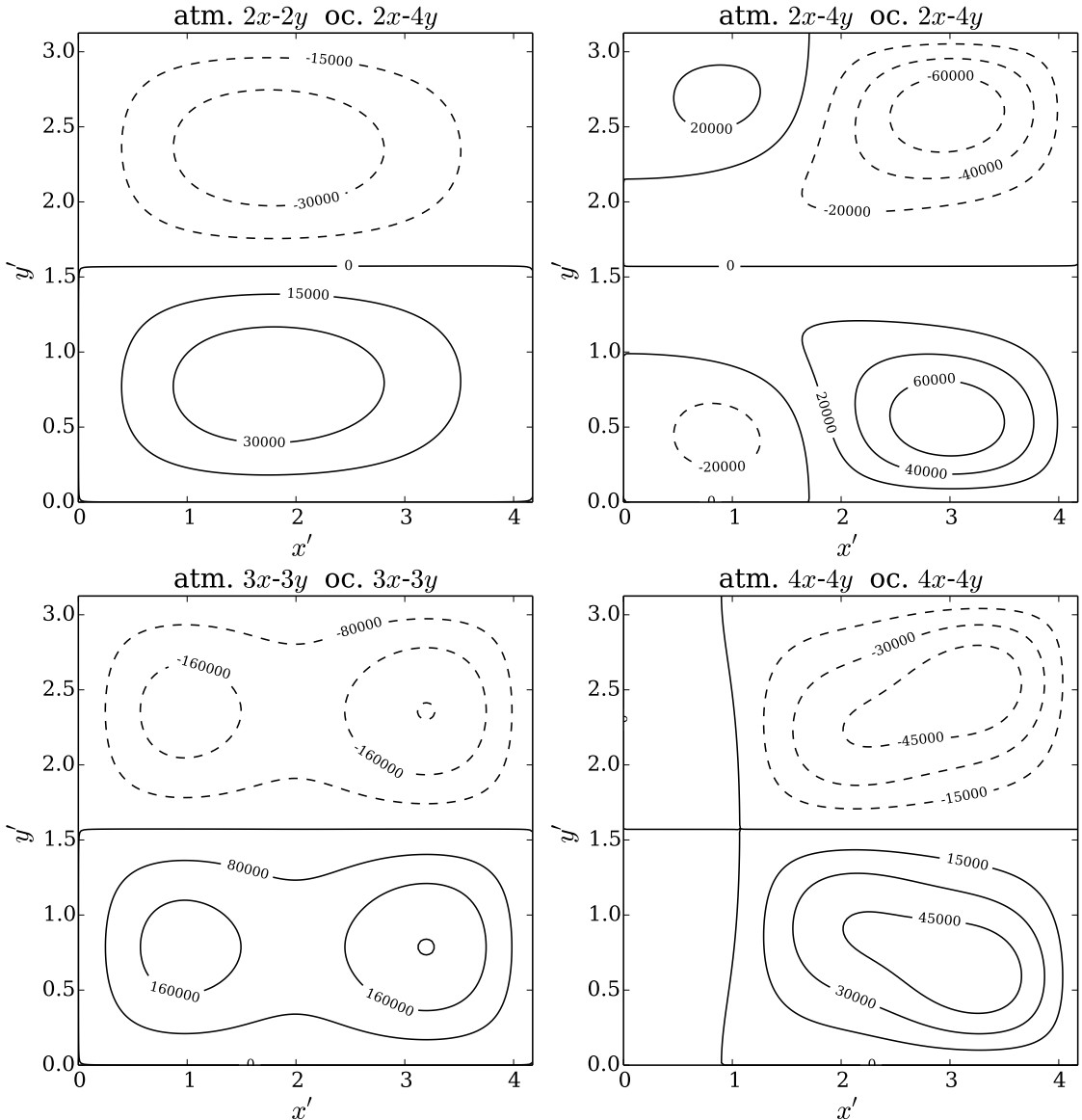

**Figure 11.** Climatologies for the oceanic streamfunction field $\psi_o$ (m$^2$ s$^{-1}$) presented on the non-dimensional model domain, as obtained using 92179.6 years of model integration.

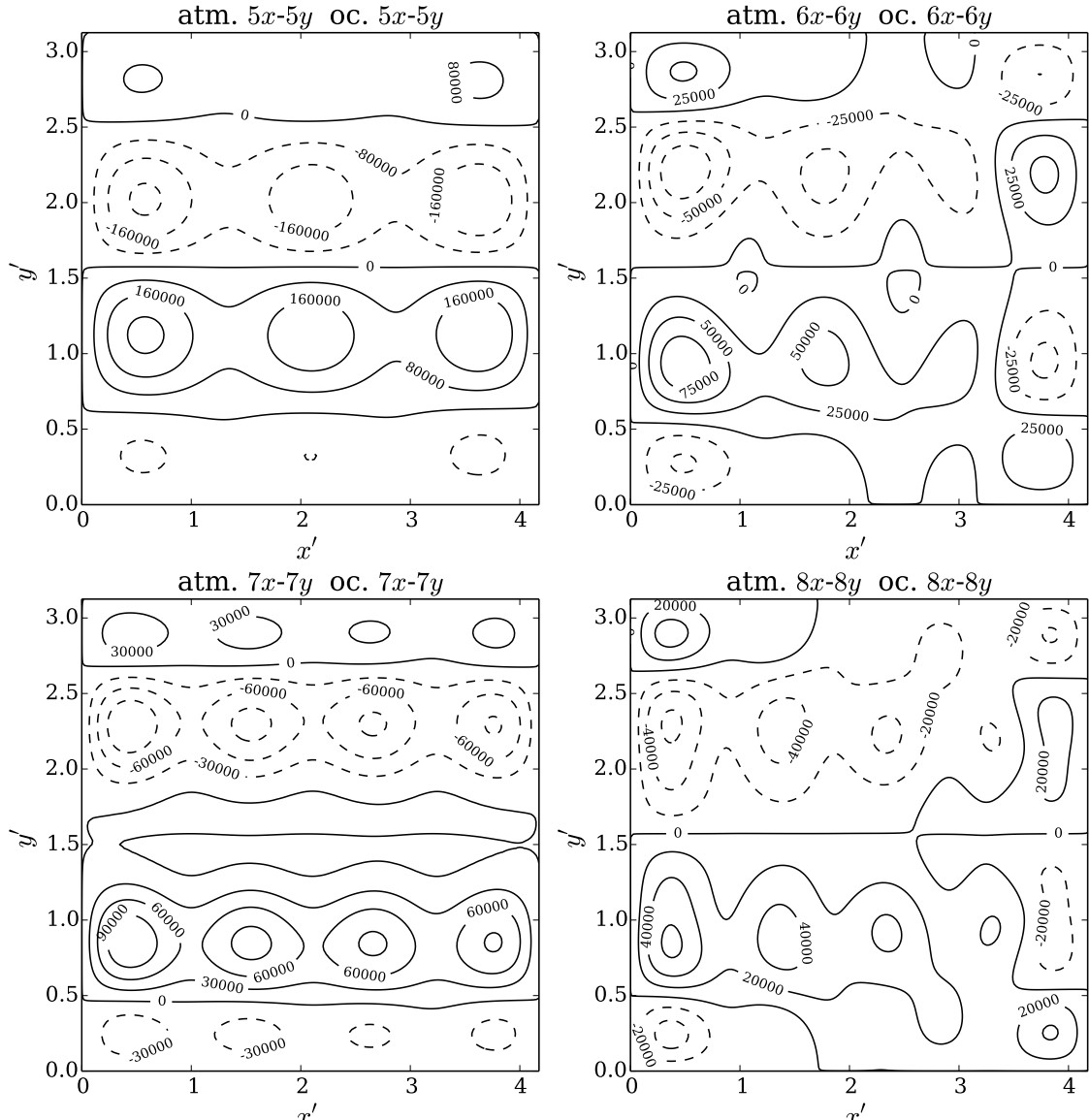

**Figure 12.** Climatologies for the oceanic streamfunction field $\psi_o$ (m$^2$ s$^{-1}$) (continued from Fig. 11).

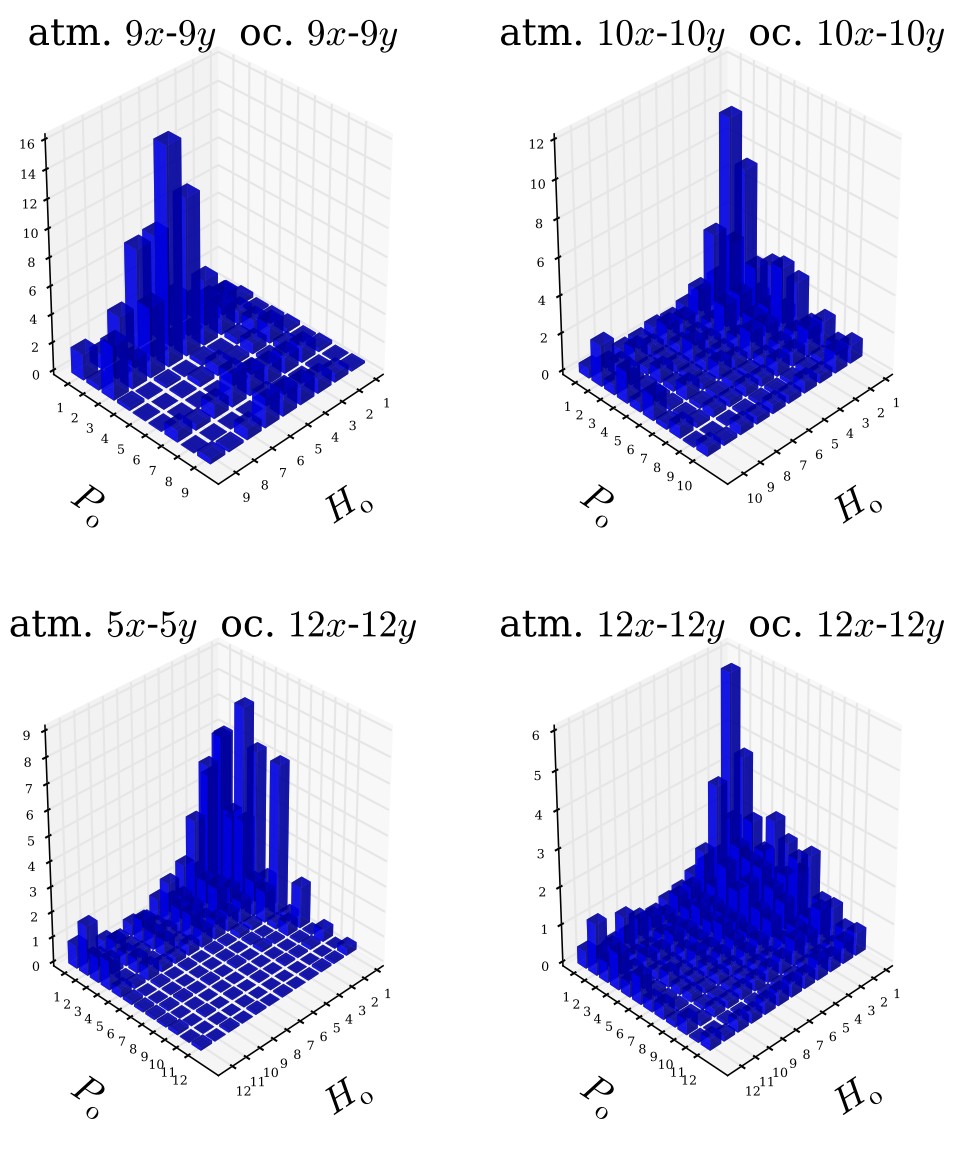

**Figure 13.** Variance distributions of the $\psi_{o,i}$ variables in percents for the high-resolution runs. The information on the different runtimes is gathered in Table 2.

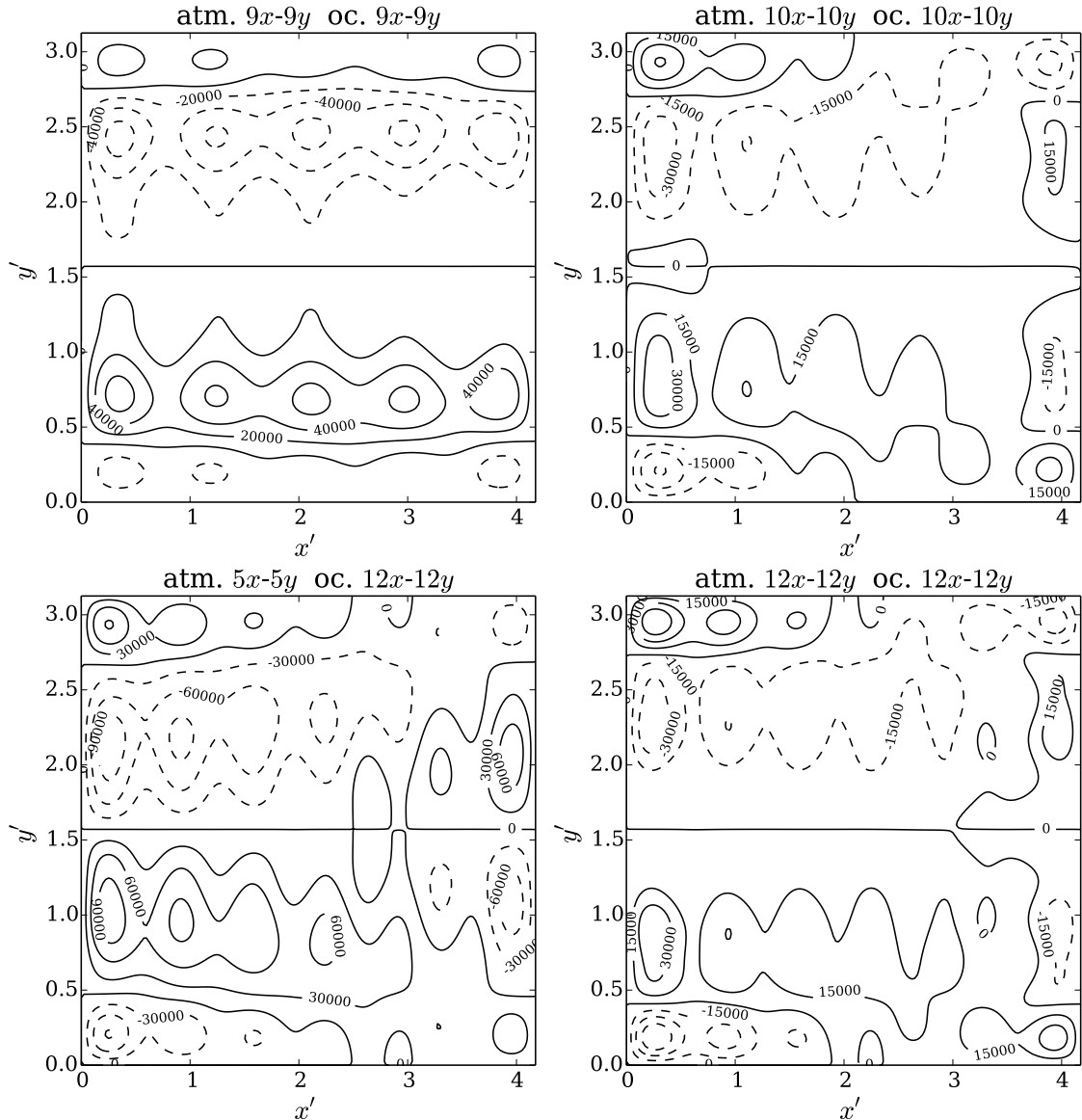

**Figure 14.** Climatologies for the oceanic streamfunction $\psi_o$ field (m$^2$ s$^{-1}$) presented on the non-dimensional model domain for the high-resolution runs displayed in Fig. 13.

| Parameter (unit) | Value | Parameter (unit) | Value |
|---|---|---|---|
| $n = 2L_y/L_x$ | 1.5 | $L_R$ (km) | 19.93 |
| $L_y = \pi L$ (km) | $5.0 \times 10^3$ | $\rho$ (kg m$^{-3}$) | 1000 |
| $f_0$ (s$^{-1}$) | $1.032 \times 10^{-3}$ | $\sigma_B$ (W m$^2$ K$^{-4}$) | $5.6 \times 5.610^{-8}$ |
| $\lambda$ (W m$^{-2}$ K$^{-1}$) | 15.06 | $\sigma$ (m$^2$ s$^{-2}$ Pa$^{-2}$) | $2.16 \times 10^{-6}$ |
| $r$ (s$^{-1}$) | $1.0 \times 10^{-7}$ | $\beta$ (m$^{-1}$s$^{-1}$) | $1.62 \times 10^{-11}$ |
| $d$ (s$^{-1}$) | $1.1 \times 10^{-7}$ | $R$ (J kg$^{-1}$ K$^{-1}$) | 287 |
| $C_o$ (W m$^{-2}$) | 310 | $\gamma_o$ (J m$^{-2}$ K$^{-1}$) | $5.46 \times 10^8$ |
| $C_a$ (W m$^{-2}$) | $C_o/3$ | $\gamma_a$ (J m$^{-2}$ K$^{-1}$) | $1.0 \times 10^7$ |
| $k_d$ (s$^{-1}$) | $3.0 \times 10^{-6}$ | $T_a^0$ (K) | 289.30 |
| $k_d'$ (s$^{-1}$) | $3.0 \times 10^{-6}$ | $T_o^0$ (K) | 301.46 |
| $h$ (m) | 136.5 | $\epsilon_a$ | 0.7 |

**Table 1.** Values of the parameters of the model that are used in the analyses of Sect. 3.

| Resolution | No. of variables | Transient (y) | Eff. runtime (y) |
|---|---|---|---|
| atm. $2x$-$2y$ oc. $2x$-$4x$ | 36 | 30726.5 | 92179.6 |
| atm. $2x$-$4y$ oc. $2x$-$4x$ | 56 | 30726.5 | 92179.6 |
| atm. $3x$-$3y$ oc. $3x$-$3x$ | 60 | 30726.5 | 92179.6 |
| atm. $4x$-$4y$ oc. $4x$-$4x$ | 104 | 30726.5 | 92179.6 |
| atm. $5x$-$5y$ oc. $5x$-$5x$ | 160 | 30726.5 | 92179.6 |
| atm. $6x$-$6y$ oc. $6x$-$6x$ | 228 | 30726.5 | 92179.6 |
| atm. $7x$-$7y$ oc. $7x$-$7x$ | 308 | 30726.5 | 92179.6 |
| atm. $8x$-$8y$ oc. $8x$-$8x$ | 400 | 30726.5 | 92179.6 |
| atm. $9x$-$9y$ oc. $9x$-$9y$ | 504 | 30726.5 | 92179.6 |
| atm. $10x$-$10$ oc. $10x$-$10y$ | 620 | 30726.5 | 92179.6 |
| atm. $5x$-$5y$ oc. $12x$-$12y$ | 398 | 15363.3 | 92179.6 |
| atm. $12x$-$12y$ oc. $12x$-$12y$ | 888 | 15363.3 | 74972.7 |

**Table 2.** Number of variables, transient time and effective runtime of the runs (in years).