# Peer review of "A Modular Arbitrary-Order Ocean-Atmosphere Model: MAOOAM v1.0"

_Geoscientific Model Development, 2016_

## Referee Comment (RC1) · Anonymous Referee #1 · 23 May 2016

General Comments: A paper titled "Modular Arbitrary-Order Ocean-Atmosphere Model: MAOOAM v1.0 " was proposed by Lesley De Cruz, Jonathan Demaeyer, and Stéphane Vannitsem.

The paper presents a new version of the VDDG model (Vannitsem et al., 2015) that allow considering an arbitrary number of modes into which dynamics may be expanded. The model was named "Modular Arbitrary-Order Ocean-Atmosphere Model" (MAOOAM) and it really seems a promising tool to investigate unresolved issues of the climate modelling. The authors analysed the dependence of the model dynamics on the truncation showing the behavior of the coupled low-frequency variability when the number of modes is increased. They finally proposed an optimal configuration for a tractable and meaningful analysis of the dynamics.

[Figure]

I think the paper is very well written. Clear, concise, incisive and the text include an appropriate number of references. Thus, I would only suggest a number of minor adjustments that, in my opinion, would improve this nice reading.

Specific Comments:

P = page; L = line;

- The section Introduction is clear as well as the presentation of the numerical treatments that is accompanied by a necessary Appendix that I think is never cited in the text; this could help the reader.

- At P1-L24: an acronym has not been declared (please check author guide if reference you provided is enough).

- At P2-L30: could you explain the origin of the values (found in table 1) you have used for interfacial friction and why they have the same value? Thank you.

- At P6-L6-10: You could reformulate the paragraph, which gives fragmented information. I would not use the word "similar"; you could briefly explain why you use Heun method. You could inform the reader about any available parallelized version somewhere else (e.g. conclusions).

- P6-L24-25: I feel it is not necessary to state this point here. - P8-L7: beta in equation 22 was not defined, please verify if it would be necessary in this context.

- P8-L19-21: what you state in this sentence could be better-understood weather you assign the same range to y-axis for all figures (at least for those that are more comparable).

- P8-L22: could you add reference or be more specific on such feature of the LFV of the NAO?

- P8-L29-30: this aspect is very interesting and it is a pity to find it sparse along the text. Why not to write a more comprehensive brief paragraph in Conclusions, stating

potential further experiments that might unveil the reason of such feature?

- P9-L28: You use the word "version" to indicate both, this new MAOOAM v1.0 versus the previous VDDG and each configuration of the model that is defined in Table 2. Am I right? I will use version in the first case and configuration in the second one.

- P9-L15-17: I think you could avoid this sentence here in two ways: by introducing MAOOAM as part of an existing model hierarchy in Introduction or by saying this in Conclusion.

- P10-L1-23: I think you could improve this part of Conclusion in order to make it clearer.

Thank you
* * *

---

## Referee Comment (RC2) · T. Sengul (Referee) · 2 Jun 2016

The paper presents a new reduced order quasi-geostrophic ocean-atmosphere model MAOOAM. The model is obtained from the quasi-geostrophic model by projecting onto the a finite dimensional space of suitably chosen basis functions.

The MAOOAM model is a generalisation of the VDDG model. The main advantages of this model to VDDG are two-fold: First, its flexibility in allowing any number of modes, second, its modularity in allowing independent selection of ocean/atmosphere modes in the zonal/meridional directions.

After the derivation of the model, the paper studies the impact of the resolution (number of modes) of the model to the properties of the attractors and the variance distributions in both the oceanic and atmospheric components.

I found the paper to be well-written and up to point. I believe the model proposed might be very useful for researchers studying the coupled ocean-atmosphere system in mid-latitude climate.

---

## Author Response (AR1)

**Author's Response to the Comments on "A Modular Arbitrary-Order Ocean-Atmosphere Model: MAOOAM v1.0"**

Lesley De Cruz, Jonathan Demaeyer, and Stéphane Vannitsem

Royal Meteorological Institute of Belgium, Avenue Circulaire 3, 1180 Brussels, Belgium

*Correspondence to:* L. De Cruz (lesley.decruz@meteo.be)

**1 Reply to the Editor**

Please find attached our response to the referees' comments on our manuscript, titled "A Modular Arbitrary-Order Ocean-Atmosphere Model: MAOOAM v1.0". This includes a point-by-point list of the modifications that we have implemented in response to the specific comments. A marked-up version of the manuscript is also included.

**2 Reply to Anonymous Referee #1**

We would like to thank Anonymous Referee #1 for their encouraging and constructive comments, which have improved the manuscript. Below is a list of modifications that we have implemented based on your specific comments.

*The section Introduction is clear as well as the presentation of the numerical treatments that is accompanied by a necessary Appendix that I think is never cited in the text; this could help the reader.*

Thank you for pointing this out. We have adapted the following paragraphs, so as to refer to the relevant sections in the Appendix.

The original paragraph (P5-L14-15) read:

"We reiterate these in Appendix A, and extend them with the formulae for both the ocean and the ocean-atmosphere coupling terms."

The new version (P5-L14-15) reads:

"We reiterate these algebraic formulae in Sect. A1 of Appendix A, and extend them with the formulae for both the ocean-atmosphere coupling terms, and the ocean inner products in Sect. A2. "

We have added the following sentence to the paragraph at (P6-L1):

"The elements of the tensor $\mathcal{T}_{i,j,k}$ are specified in Appendix B."

*At P1-L24: an acronym has not been declared (please check author guide if reference you provided is enough).*

Indeed. The sentence:

"The first of these, OA-QG-WS v1 (Vannitsem, 2014), features only mechanical coupling between the ocean and the atmosphere, and uses 12 atmospheric variables following Charney and Straus (1980) and four oceanic modes following Pierini (2011)."

has been changed to (P1-L23-24):

"The first of these, OA-QG-WS v1 (Vannitsem, 2014), for Ocean-Atmosphere - Quasi-Geostrophic - Wind Stress, features only mechanical coupling between the ocean and the atmosphere, and uses 12 atmospheric variables following Charney and Straus (1980) and four oceanic modes following Pierini (2011)."

*At P2-L30: could you explain the origin of the values (found in table 1) you have used for interfacial friction and why they have the same value? Thank you.*

The estimation of the parameters $k_d$ and $k_d'$ as well as other parameters is detailed in Vannitsem (2015). In Nese and Dutton (1993) the (non-dimensionalized) parameter $k$ is estimated to lie within [0.1,0.01]. Our choice of $k_d = k_d' = 3 \times 10^{-6}$ corresponds to a value of $k = 0.0145$ and $k' = 2k = 0.0290$, with $k_d = 2k\, f_0$ and $k_d' = k\, f_0$. The parameters $k_d$ and $k_d'$ are chosen to have the same value, as was done in Charney and Straus (1980).

The sentence on P7-L3 originally read:

"These were selected so as to lie within the realistic bounds derived in Vannitsem (2015)."

This has been corrected as follows (P7-L3-5 in the new version):

"The parameter values for $L$, $L_R$, $\lambda$, $r$, $d$, $C_o$, $C_a$, $k_d$ and $k_d'$ were selected as detailed in Vannitsem et al. (2015). The same value was chosen for $k_d$ and $k_d'$, as was done in Charney and Strauss (1980), see also Vannitsem and De Cruz (2014)."

*At P6-L6-10: You could reformulate the paragraph, which gives fragmented information. I would not use the word "similar"; you could briefly explain why you use Heun method. You could inform the reader about any available parallelized version somewhere else (e.g. conclusions).*

Thank you for your suggestion. We have reformulated the original paragraph:

"Two implementations of MAOOAM are provided as supplementary material: one in Lua and one in Fortran. The Lua code is optimized for LuaJIT, a just-in-time compiler for Lua (Pall, 2015). The performance of both implementations is similar. The model equations are numerically integrated using the Heun method, but one can choose a different integration method; as an example, a fourth-order Runge-Kutta integrator is also included in the Lua implementation. This implementation is also available in a parallelized version, which uses MPI."

We have quantified the "similar" performance, and clarified the use of the Heun method. The paragraph now reads (P6-L6-10 in the new version):

"Two implementations of MAOOAM are provided as supplementary material: one in Lua and one in Fortran. The Lua code is optimized for LuaJIT, a just-in-time compiler for Lua (Pall, 2015), and runs about 20% slower than the Fortran version. By default, the model equations are numerically integrated using the Heun method. We have tested higher-accuracy methods, but these did not significantly change the results. The integration method can easily be changed; as an example, a fourth-order Runge-Kutta integrator is also included in the Lua implementation."

Furthermore, we have moved the information on the parallelized version to Sect. 5: Code availability. The paragraph read:

"MAOOAM v1.0 is freely available for research purposes in the supplementary material and is also available at http://github.com/Climdyn/MAOOAM. In addition, the code is archived at http://dx.doi.org/10.5281/zenodo.47507 and http://dx.doi.org/10.5281/zenodo.47510 (parallel version)."

This has been extended to (P10-L32 to P11-L2 in the new version):

"MAOOAM v1.0 is freely available for research purposes in the supplementary material and is also available at http://github.com/Climdyn/MAOOAM. In addition, the code is archived at http://dx.doi.org/10.5281/zenodo.47507. A version of the Lua implementation which is parallelized using MPI is also available at http://github.com/Climdyn/MAOOAM/tree/mpi. The parallelized version is archived at http://dx.doi.org/10.5281/zenodo.47510."

*P6-L24-25: I feel it is not necessary to state this point here.*

We have removed the sentence: "The ease with which these quantities can be computed makes MAOOAM useful for applications in data assimilation, sensitivity analyses and predictability studies."

*P8-L7: beta in equation 22 was not defined, please verify if it would be necessary in this context.*

It is the same beta as in Eqs. (1)-(3), but given the different context, we have repeated the definition for the sake of clarity (P8-L10 of the new version): " and $\beta = df/dy$ is the meridional derivative of the Coriolis parameter $f$."

*P8-L19-21: what you state in this sentence could be better-understood weather you assign the same range to y-axis for all figures (at least for those that are more comparable).*

We have adapted the $y$-axis for all plots in Figs. 7 and 8, except for the lowest model resolution, which has a significantly wider range.

*P8-L22: could you add reference or be more specific on such feature of the LFV of the NAO?*

We have rephrased this more clearly, and added the relevant references. The original sentence read:

"Moreover, at high resolution this LFV is weaker, which seems closer to reality as revealed, for instance, by the weak LFV of the NAO index."

The new sentence reads (P8-L23-25):

"Moreover, at high resolution this LFV is weaker than for the VDDG model version, but it seems closer to the actual dynamics found for the North-Atlantic Oscillation (NAO) as discussed in Li and Wang (2003) and Stephenson et al. (2000)."

*P8-L29-30: this aspect is very interesting and it is a pity to find it sparse along the text. Why not to write a more comprehensive brief paragraph in Conclusions, stating potential further experiments that might unveil the reason of such feature?*

Following your suggestion, we have removed the following sentence from Sect. 3:

"Note also that, at higher resolution, the structure of the LFV seems to depend on whether $H^{\max}$, $M^{\max}$, $H_o^{\max}$ and $P_o^{\max}$, $P^{\max}$ are even or odd numbers."

Instead, we have added this paragraph to the conclusion (P10-L22-26 of the new version):

"Another interesting finding is the change of structure of the climatologies of the ocean gyres when choosing even or odd wave numbers ($H^{\max}$, $M^{\max}$, $H_o^{\max}$ and $P_o^{\max}$, $P^{\max}$). Is this feature purely associated with the convergence toward a spatially continuous field, or does it reflect specific properties of the dynamical equations, such as symmetries or invariance?

These questions are still open and will be the subject of a future investigation that should allow to clarify what is the best set of modes needed for the ocean description."

*P9-L28: You use the word "version" to indicate both, this new MAOOAM v1.0 versus the previous VDDG and each configuration of the model that is defined in Table 2. Am I right? I will use version in the first case and configuration in the second one.*

Thank you for pointing this out. We have replaced the ambiguous word "version" by "configuration" throughout the manuscript, where needed.

*P9-L15-17: I think you could avoid this sentence here in two ways: by introducing MAOOAM as part of an existing model hierarchy in Introduction or by saying this in Conclusion.*

We have removed the following sentence from Sect. 3 (P9-L15-17 in the old version):

"It must however be stressed that the VDDG model is still an important tool in this hierarchy of models since it already contains the basic mechanisms leading to low-frequency variability."

The following sentence has been added to the Conclusion (P10,L20-21 in the new version):

"Note that the VDDG model is still an important tool in this hierarchy of models, since it already contains the basic mechanisms leading to the LFV."

*P10-L1-23: I think you could improve this part of Conclusion in order to make it clearer.*

We have restructured and added content in the Conclusion, following your suggestions above. Thank you for these valuable comments. In addition, we have clarified the sentence (P10-L5-6):

"Consequently, none of the solutions presented so far have satisfactorily converged toward a dynamics that correctly reflects the wave-dominated behaviour of the coupled ocean-atmosphere system. "

The new version now reads (P10-L7-10):

"Consequently, none of the solutions presented so far have satisfactorily converged toward a dynamics that correctly reflects the wave-dominated regime of the coupled ocean-atmosphere system. This regime corresponds to a resolution associated with the Rhines scale (which for the ocean is equal to 100 km, or equivalently, to wavenumbers of the order of $H_o^{\mathrm{max}}/2 \approx P_o^{\mathrm{max}} \approx 50$)."

A version of the manuscript is provided, in which all changes are highlighted.

**3   Reply to the second referee, Dr. Taylan Sengul**

We thank Dr. Sengul for his positive and supportive review. It is our hope that the model will indeed prove useful to the scientific community and allow for a better understanding of the ocean-atmosphere coupling.

**4   Manuscript version with highlighted changes**

[revised manuscript text omitted]
^{\mathrm{max}}$, $M^{\mathrm{max}}$, $H_o^{\mathrm{max}}$ and $P_o^{\mathrm{max}}$, $P^{\mathrm{max}}$ are even or odd numbers.

The previous results point toward the important question of the optimal resolution of the oceanic component needed to get a sufficiently  low-resolution model, while keeping a dynamics with strong similarities with a very
5  high-resolution model. To answer this question, we have performed some higher resolution integrations, but on shorter time spans. The time span for each integration is given in Table 2.

The variance distributions of the oceanic streamfunction variables (see Fig. 13) have decreased at the spectral cut-off's edges compared to the distributions of the lower resolution model  configurations shown on Fig. 5. However, this decrease is not sufficient and apparently spurious effects are still present. For instance, the decay is not identical in both directions, with a
10  slower decay rate as the zonal wave-number $H_o$ increases. We can even notice a peak in the distribution around $H_0 = H_0^{\mathrm{max}}$ and $P_0 = 2$ for all these higher model resolutions. This indicates that in fact we are still far from a quantitatively representative solution in the ocean. It supports that, as stated previously, a resolution of the order of the Rhines scale is needed to achieve a good convergence. For the ocean, it corresponds to a 100 km resolution which would then require roughly 2000 modes. Such a model will of course be very computationally expensive and cannot be considered as a "reduced"-order model anymore.
15  However, the comparison between the atm. $5x$-$5y$ oc. $12x$-$12y$ model  configuration and the $10x$-$10y$ or the $12x$-$12y$ model  configuration shows that the former displays a large-scale behaviour close to the latter two, but with a reduced complexity and computational cost. This similarity can be assessed by considering the climatologies of these higher-resolution runs displayed in Fig. 14 and by watching the corresponding videos (see below). We therefore believe that the atm. $5x$-$5y$ oc. $12x$-$12y$ model  configuration is a good candidate when investigating more realistic dynamics than the one presented in
20  VDDG. It must however be stressed that the VDDG model is still an important tool in this hierarchy of models since it already contains the basic mechanisms leading to low-frequency variability. In addition, the climatologies shown in Fig. 14 confirm the dependence of the dynamics on whether $H^{\mathrm{max}}$, $M^{\mathrm{max}}$, $H_o^{\mathrm{max}}$ and $P_o^{\mathrm{max}}$, $P^{\mathrm{
[revised manuscript text omitted]